# Target guided synthesis using DNA nano-templates for selectively assembling a G-quadruplex binding *c-MYC* inhibitor

Deepanjan Panda[1], Puja Saha[1], Tania Das[1] & Jyotirmayee Dash[1]

The development of small molecules is essential to modulate the cellular functions of biological targets in living system. Target Guided Synthesis (TGS) approaches have been used for the identification of potent small molecules for biological targets. We herein demonstrate an innovative example of TGS using DNA nano-templates that promote Huisgen cycloaddition from an array of azide and alkyne fragments. A G-quadruplex and a control duplex DNA nano-template have been prepared by assembling the DNA structures on gold-coated magnetic nanoparticles. The DNA nano-templates facilitate the regioselective formation of 1,4-substituted triazole products, which are easily isolated by magnetic decantation. The G-quadruplex nano-template can be easily recovered and reused for five reaction cycles. The major triazole product, generated by the G-quadruplex inhibits *c-MYC* expression by directly targeting the *c-MYC* promoter G-quadruplex. This work highlights that the nano-TGS approach may serve as a valuable strategy to generate target-selective ligands for drug discovery.

---

[1] Department of Organic Chemistry, Indian Association for the Cultivation of Science, Jadavpur, Kolkata-700032, India. Correspondence and requests for materials should be addressed to J.D. (email: ocjd@iacs.res.in).

Small molecules are fundamental probes for studying biological system. Considering the dynamic nature of the biological macromolecules, the development of small molecules that can interact with biological targets with high affinity and selectivity has always been challenging and time consuming. Target Guided Synthesis (TGS) is a powerful approach to discover novel and selective small molecule binders for biological targets by combining the synthesis and screening into a single step[1]. It uses a biological target to synthesize its own best binders from a series of small molecule fragments, functionalized with complementary reacting groups. In a pioneering work, Sharpless and co-workers developed *in situ* click chemistry, a kinetically controlled TGS approach in which the enzyme acetylcholinesterase was used as the target to assemble its potent small molecule inhibitors from a pool of azide and alkyne building blocks[2,3]. In TGS, the azide and alkyne fragments undergo Huisgen 1,3-dipolar cycloaddition (Click reaction) in the presence of a catalytic target that brings the azide and alkyne building blocks into close proximity with correct spatial orientation to generate one of the triazole regioisomers in the absence of any metal catalyst. Therefore, small molecules identified by TGS are expected to show high binding affinity as well as specificity for the target as they are synthesized by a specific reaction in which the active site of the biological target controls the assembly of the best binding fragments. The majority of TGS approaches, reported to date, use various enzymes as the target to assemble their potent inhibitors[4–7]. Only two examples are reported so far using nucleic acids as the targets[8,9]. However, these solution phase TGS methods have some limitations like poor isolation of the lead compounds from the reaction mixture comprising the target and fragment library and lack of reusability of the target for multiple rounds of templated synthesis.

We anticipated that the use of magnetic nanoparticle supported DNA sequences as the targets could overcome such shortcomings; the *in situ* lead compounds could be easily isolated by simple magnetic decantation and the DNA templates could be easily recovered and recycled. In addition, the DNA sequences immobilized on the nanoparticle surface would greatly enhance the rate of reaction between small molecule fragments by providing a larger surface area with more numbers of localized DNA templates.

In the present study, we describe an azide-alkyne cycloaddition based TGS approach using *c-MYC* G-quadruplex and control duplex DNA monolayers assembled on gold-coated magnetic-nanoparticles as the templates. The *c-MYC* G-quadruplex present in the NHE III1 region upstream of the P1 promoter is believed to be involved in the complex regulation of *c-MYC* expression and thus considered as an attractive target for anticancer therapeutics[10–15]. Small molecules capable of binding *c-MYC* G-quadruplex have been developed using multistep organic syntheses[16–19]. Although many ligands are known to stabilize G-quadruplexes and modulate gene expression, only recently a benzofuran derivative is reported to inhibit *c-MYC* expression by G-quadruplex dependent mechanism[20]. Our findings demonstrate that the G-quadruplex nano-template can be efficiently used in TGS for the rapid identification of a selective triazole ligand for the *c-MYC* G-quadruplex. Using biophysical and cellular assays, we also establish that this compound can inhibit the *c-MYC* expression via G-quadruplex binding.

## Results

### Preparation of DNA functionalized Au@Fe$_3$O$_4$ nanoparticles.
To develop TGS using DNA nano-templates, the thiolated DNA sequences were immobilized on the surface of gold-coated magnetic nanoparticles (Au@Fe$_3$O$_4$). Iron oxide based magnetic nanoparticles (Fe$_3$O$_4$ MNPs) have shown good promise in biological applications as they are non-toxic, naturally available and easily synthesizable with superior magnetic and chemical properties[21–26]. Gold coating on these nanoparticles improves the stability of the nanoparticles in solution, makes them biocompatible and provides an outer layer for covalently tethering thiolated biomolecules on nanoparticle surfaces by chemisorption. The Au@Fe$_3$O$_4$ nanoparticles were prepared by citrate reduction of gold salt on Fe$_3$O$_4$ nanoparticles (see Methods)[27].

The thiolated G-quadruplex and duplex DNA sequences were self-assembled as a monolayer by covalent bonding with the surface of the gold shell (Fig. 1a)[28], which is confirmed by the ultraviolet-visible absorption spectra (Fig. 1b). The Fe$_3$O$_4$ MNP spectrum showed no characteristic peak in the visible region, while the dispersion of Au@Fe$_3$O$_4$ NPs in 100 mM Tris-KCl buffer (pH 7.4) displayed a SPR absorption band ∼535 nm (refs 29–31). The DNA linked Au@Fe$_3$O$_4$ NPs (G$_4$•Au@Fe$_3$O$_4$ and *ds*DNA•Au@Fe$_3$O$_4$) also showed characteristic SPR bands of Au at 535 nm. The G$_4$•Au@Fe$_3$O$_4$ and *ds*DNA•Au@Fe$_3$O$_4$ displayed absorption peaks near 270 and 280 nm, respectively, suggesting DNA modification on the MNP surface (Fig. 1b). The surface functionalization of the nanoparticles with thiolated *c-MYC* G$_4$ DNA was further confirmed by CD spectroscopy. The CD spectrum of G$_4$•Au@Fe$_3$O$_4$ nano-template exhibited a positive peak at 260 nm and a negative peak at 240 nm, which indicates that it retains the parallel conformation of the *c-MYC* quadruplex (Fig. 1c). The DNA-linked nanoparticles exhibited paramagnetic properties of Fe$_3$O$_4$ MNPs (Supplementary Fig. 1). Then we characterized the DNA-linked nanoparticles by transmission electron microscopy (TEM) and atomic force microscopy (AFM) experiments. Both the Fe$_3$O$_4$ and Au@Fe$_3$O$_4$ were nearly spherical and relatively mono-disperse with a mean diameter of 11 and 15 nm, respectively, (Supplementary Fig. 1). The EDX spectra of Au@Fe$_3$O$_4$ NP indicated the presence of both Fe and Au in the particles (Supplementary Fig. 1). The G$_4$•Au@Fe$_3$O$_4$ MNPs also preserved the near spherical shape of the NPs (Fig. 1d). The attachment of thiol-DNA onto Au@Fe$_3$O$_4$ MNPs was further assessed by β-mercaptoethanol (β-ME) mediated thiol exchange reaction that displaced the surface bound *c-MYC* G$_4$ DNA, visualized by Native PAGE analysis (Supplementary Fig. 2)[32].

### Design and synthesis of azide and alkyne building blocks.
To generate high-affinity G-quadruplex ligands, a library of azide and alkyne building blocks was prepared (Fig. 2). The alkynes **1a**, **1b** and **1c** were derived from carbazole, a rigid heteroaromatic ring system present in numerous natural products and pharmacologically active compounds[33–36]. The alkyne **1a** was prepared in good overall yield from the commercially available carbazole using mono-iodination, N-arylation followed by amide coupling and Sonogoshira reaction (Supplementary Fig. 3). Alkyne **1b** was simply prepared from the mono-iodo carbazole by Sonogashira coupling (Supplementary Fig. 4) and alkyne **1c** was derived from mono-iodo carbazole by N-alkylation followed by Sonogoshira coupling (Supplementary Fig. 5). The reason for selecting carbazole ring system is that it can interact with the G-quadruplex DNA through π–π stacking interactions with the external G-quartets. Our azide library contains a range of aliphatic and aromatic azides that includes amines (**2**, **3** and **7**), alcohols (**4** and **7**), simple phenyl ring (**5**), aromatic aldehyde (**8**), ester (**9**) and carboxamide with amine side chains (**11**). It also contains amino acid and nucleoside derivatives such as prolinamide (**10**) and guanosine (**12**). These azide building blocks (**2–12**) may interact with G-quadruplex using π–π stacking interactions by the aromatic ring systems and electrostatic

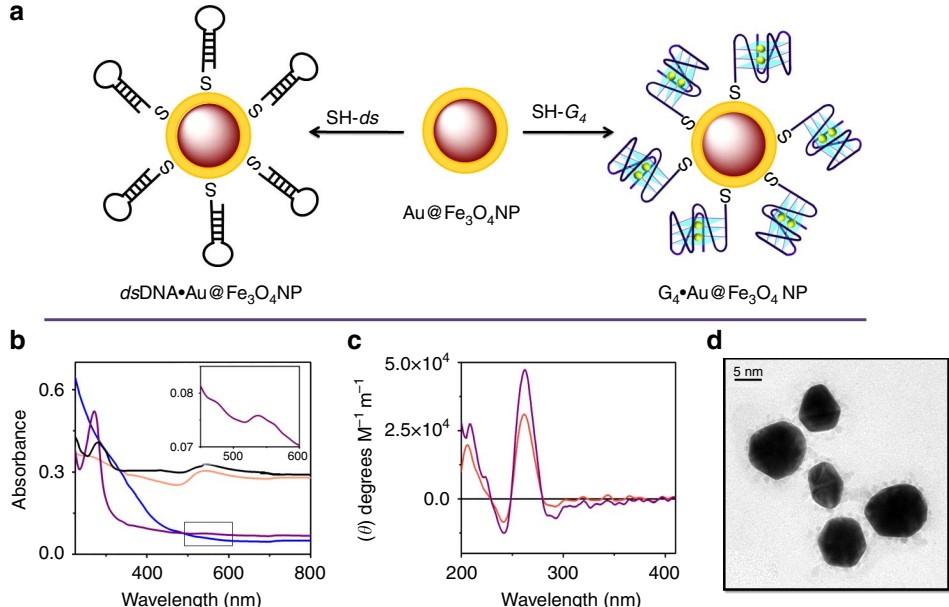

**Figure 1 | Characterization of G$_4$•Au@Fe$_3$O$_4$ NPs.** (**a**) Schematic diagram of formation of *c-MYC* G-quadruplex (*5'-SH-G$_4$AG$_3$TG$_4$AG$_3$TG$_4$-3'*) and duplex DNA (*5'-SH-CA$_5$T$_5$GCA$_5$T$_5$G-3'*) functionalized Au@Fe$_3$O$_4$ nanoparticles. (**b**) Absorption spectra of magnetic nanoparticles in Tris-KCl buffer (pH 7.4); blue-Fe$_3$O$_4$ NP, orange-Au@Fe$_3$O$_4$ NP, violet-G$_4$•Au@Fe$_3$O$_4$ and black- *ds*DNA•Au@Fe$_3$O$_4$ NP. (**c**) CD spectrum of the thiol-capped *c-MYC* G-quadruplex oligonucleotide bound nanoparticles (G$_4$•Au@Fe$_3$O$_4$); magenta-10 μM *c-MYC* G-quadruplex DNA, orange-G$_4$•Au@Fe$_3$O$_4$. (**d**) TEM image of G$_4$•Au@Fe$_3$O$_4$.

interactions by the protonable side chains. The synthesized azide and alkyne library would generate 66 theoretically possible triazole products including 1,4 and 1,5 regioisomers that may interact with the target G-quadruplex.

**G$_4$•Au@Fe$_3$O$_4$ MNPs templated synthesis.** We then evaluated the ability of G$_4$•Au@Fe$_3$O$_4$ to template the metal free 1,3-dipolar cycloaddition. In a typical experiment, a 4:1 mixture of azide and alkyne building blocks **1–12** were stirred with *c-MYC* G$_4$•Au@Fe$_3$O$_4$ NPs in Tris-KCl buffer (100 mM, pH 7.4). After 6 days, the reaction mixture was treated with the denaturing agent lithium chloride (LiCl) and heated to 65 °C to unfold the DNA-quadruplex (Supplementary Fig. 6). The HPLC chromatogram of the supernatant collected after magnetic separation of DNA•NPs showed an inseparable complex mixture of compounds with overlapping peaks (Supplementary Fig. 6). Subsequently, the purification protocol was modified by simple heating the mixture at 65 °C without adding LiCl (ref. 37). However, the *in situ* lead compounds could not be identified as overlapping peaks were obtained in HPLC (Supplementary Fig. 6).

We then modified the separation technique to easily identify and isolate the *in situ* leads. As the resultant triazole products are expected to bind the G-quadruplex magnetic nano-template, the separation technique was improved to (i) detect each generated triazole leads, enabling the analysis of the complex reaction mixture in an optimum time and (ii) recycle the DNA template for successive azide-alkyne cycloadditions. After incubating the mixture of azide and alkyne fragments **1–12** with the G$_4$•Au@Fe$_3$O$_4$ MNPs for 6 days, the MNPs along with the newly formed bound products were separated from the reaction mixture using an external magnet and washed three times with 100 mM Tris-KCl buffer to remove the unreacted building blocks. The unreacted fragments remained in the supernatant. The MNPs were then resuspended in 100 mM Tris-KCl buffer, heated to 65 °C to release the products bound with the G-quadruplex nano-template. The DNA-MNPs were separated instantly by magnetic decantation and the supernatant was collected and characterized by HPLC and ESI-MS analysis (Fig. 3, Supplementary Methods).

We observed the formation of three triazole compounds (**Tz 1–3** in 62:7:31 percentage ratio) in the presence of the *c-MYC* G$_4$•Au@Fe$_3$O$_4$ NPs (Fig. 4a). For a comparison, we have performed the TGS reaction with conventional *c-MYC* G-quadruplex DNA sequence (*5'-G$_4$AG$_3$TG$_4$AG$_3$TG$_4$-3'*), that yielded essentially similar lead compounds which were identified in ESI-MS analysis but overlapping peaks were obtained in HPLC chromatogram. This result validates the use of G$_4$•Au@Fe$_3$O$_4$ MNPs as an alternative template to the homogeneous G-quadruplex DNA for the templated synthesis of G-quadruplex ligands. This method thus provides advantages over the conventional solution phase TGS as the resulting products can be efficiently analysed and separated.

Simultaneously, the reaction was performed in the presence of MNPs functionalized with duplex DNA (*ds*DNA•Au@Fe$_3$O$_4$) (Supplementary Fig. 8) as the control and only one triazole compound **Tz 2** (combination of alkyne **1a** and azide **3**) was detected in MS-HPLC analysis (Fig. 4b). This protocol thus validates products **Tz 1** (alkyne **1a** and azide **11**) and **Tz 3** (alkyne **1a** and azide **7**) as the selective *in situ* lead compounds for the G-quadruplex DNA target (Fig. 4c,d). Compound **Tz 2** non-specifically binds to both quadruplex and duplex DNA. It is worth noting that all the lead compounds are derived from alkyne **1a**, which indicates that the DNA templates preferentially select alkyne **1a** over alkyne **1b** and **1c**. However, the solubility of these fragments in aqueous buffer may play a role in their selection. Other possible products were not formed or detected in considerable amount indicating that the corresponding azides may not be optimally oriented by the *c-MYC* G-quadruplex template to react with the alkyne partners. Moreover, the preferential formation of **Tz 1** suggests that the *c-MYC* G-quadruplex selected the azide **11** due to its aromatic carboxamide group and positively charged amine side chain that facilitates π–π stacking with additional electrostatic interactions with the quadruplex DNA.

We then analysed the product distribution of the reaction mixture with G$_4$•Au@Fe$_3$O$_4$ template at two different time points. By stirring the reaction for 2 days, the major lead **Tz 1** and

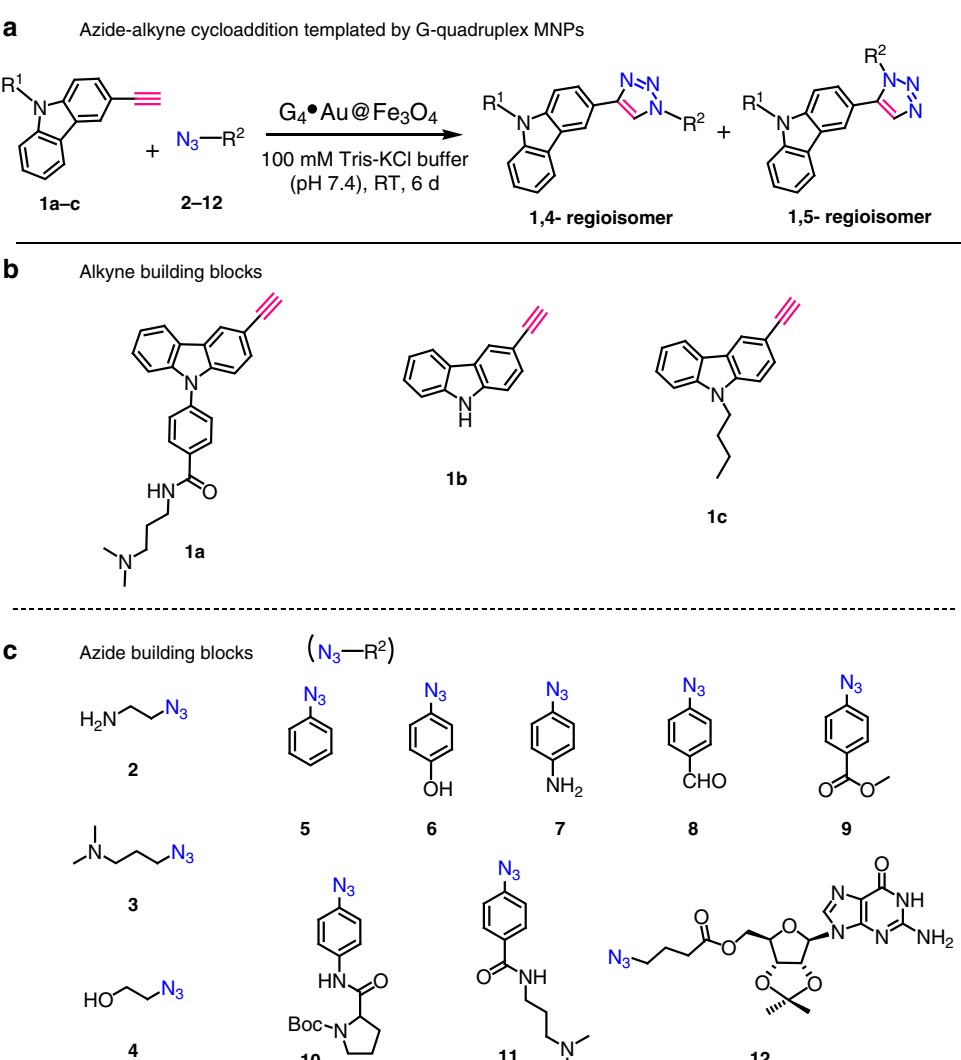

**Figure 2 | Design of building blocks for templated cycloaddition. (a)** $G_4\bullet Au@Fe_3O_4$ NP templated azide-alkyne cycloaddition. **(b,c)** Structures of carbazole alkynes **1a**–**c** and azides **2**–**12**.

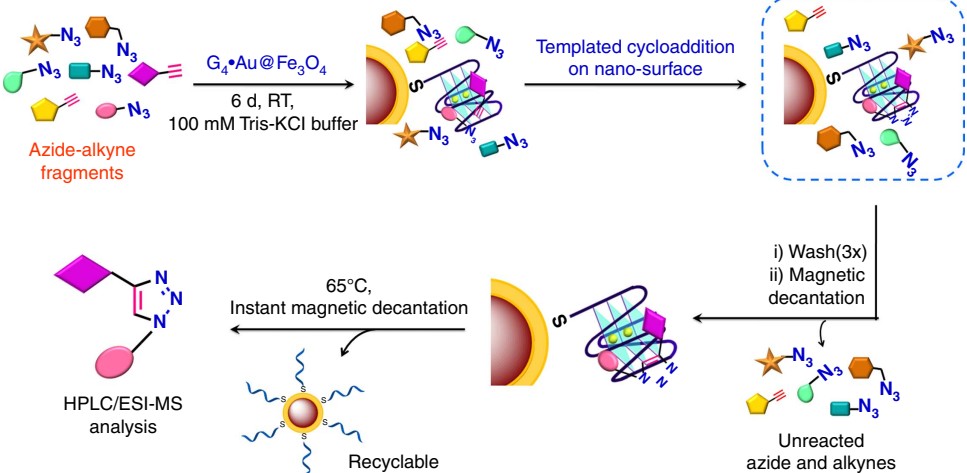

**Figure 3 | Selective cycloaddition by $G_4\bullet Au@Fe_3O_4$ nano-template.** Schematic representation of templated cycloaddition between selective azide and alkyne partners using $G_4\bullet Au@Fe_3O_4$ nanoparticles.

**Tz 2** (exclusively formed by duplex DNA) were obtained in a percentage ratio of 65:35 (Supplementary Fig. 7). After 4 days, the formation of **Tz 1** was increased compared to **Tz 2** (78:22), while the other lead **Tz 3** was not detected in the mixture (Supplementary Fig. 7). The formation of **Tz 2** was not significantly increased by the G-quadruplex template with time suggesting

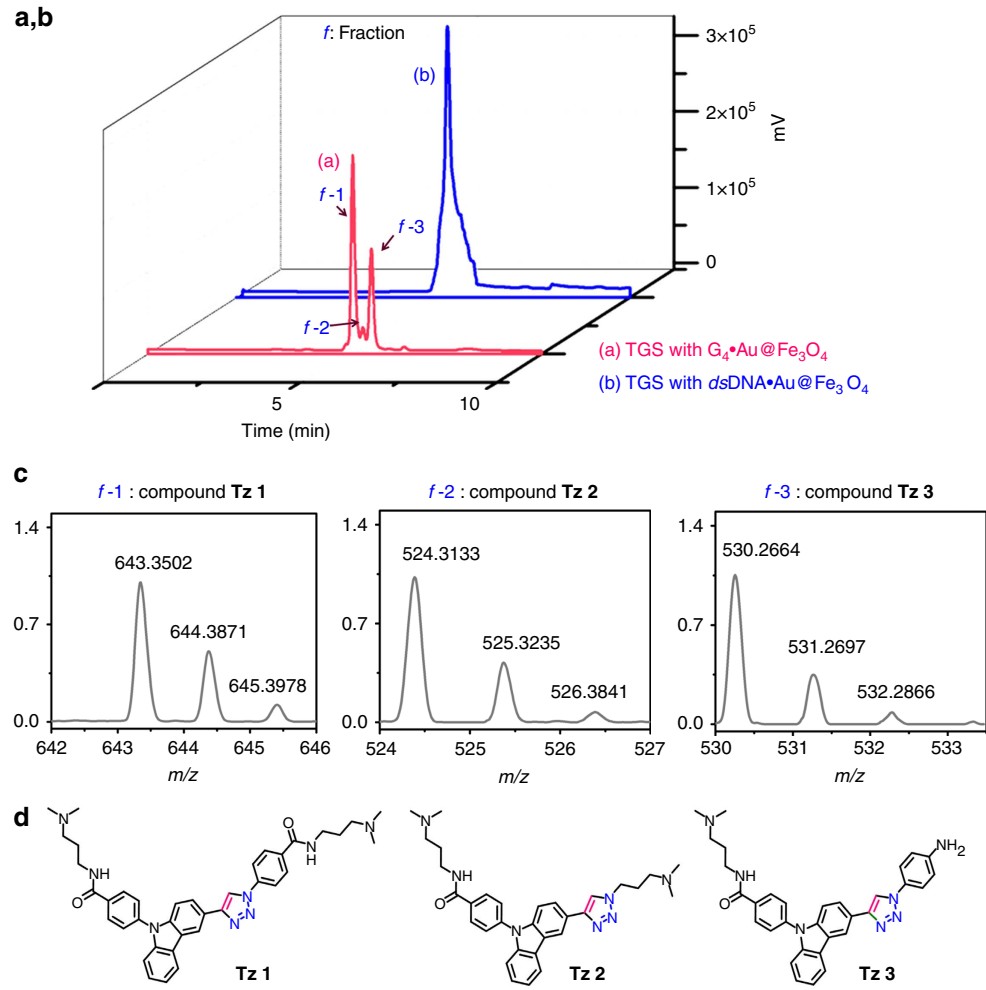

**Figure 4 | Monitoring of $G_4 \bullet Au@Fe_3O_4$ mediated target guided synthesis.** HPLC chromatogram of the supernatant collected following the modified separation protocol; HPLC analysis using (**a**) $G_4 \bullet Au@Fe_3O_4$ MNPs and (**b**) $ds$DNA$\bullet Au@Fe_3O_4$. (**c**) ESI-MS analysis of the three fractions collected from the HPLC analysis using $G_4 \bullet Au@Fe_3O_4$ reveals the formation of compounds **Tz 1**, **Tz 2** and **Tz 3**. (**d**) The molecular structures of compounds **Tz 1**, **Tz 2** and **Tz 3**.

weaker binding affinity of **Tz 2** for the *c-MYC* G-quadruplex DNA. However, the formation of **Tz 1** was greatly accelerated by $G_4$ DNA template with increasing time. These observations lead us to conclude that the *c-MYC* $G_4 \bullet Au@Fe_3O_4$ nano-template promoted the preferential formation of **Tz 1** as the potent lead compound and it could not bind other weak alkyne-azide partners on short time scales.

To determine the regioselectivity of the triazole product **Tz 1** formed by *c-MYC* $G_4 \bullet Au@Fe_3O_4$ NP templated synthesis, we have carried out the cycloaddition of alkyne **1a** and azide **11** under both thermal and Cu(I) catalysed conditions (Supplementary Fig. 9). The Cu(I) catalysed cycloaddition gave the 1,4-substituted triazole product and the thermal Huisgen cycloaddition between the corresponding azide and alkyne generated both the 1,4- and 1,5-substituted regioisomers. Both the compounds have identical mass and the 1,5-substituted triazole product has a slightly lower retention time. By comparing the HPLC retention times, we have observed that the *c-MYC* $G_4 \bullet Au@Fe_3O_4$ NPs templated the 1,3-dipolar cycloaddition of alkyne **1a** and azide **11** leading to the formation of the identical product as in the Cu(I) catalysed reaction (Supplementary Fig. 10). These results illustrate that the *c-MYC* $G_4 \bullet Au@Fe_3O_4$ NPs promote the cycloaddition between the proximally oriented alkyne **1a** and azide **11** in a regioselective manner resulting in the formation of the 1,4-regioisomer of **Tz 1** (relative yield 44%; Supplementary Fig. 11).

**Recovery and reuse of the *c-MYC* DNA nano-template**. Next, we investigated the ability of the magnetically recovered *c-MYC* $G_4 \bullet Au@Fe_3O_4$ nanoparticles to template the Huisgen cycloaddition using the same azide-alkyne fragments **1–12**. After 6 days, the recovered *c-MYC* MNPs provided the triazole products **Tz 1–3** with nearly similar product distribution (63:6:31) as the freshly prepared *c-MYC* $G_4 \bullet$MNPs (Supplementary Fig. 12). The product distribution was found to be similar in the subsequent reaction cycle. In the third cycle, **Tz 1** and **Tz 2** were obtained while the formation of **Tz 3** (the other G4-binding ligand) was not observed in the HPLC chromatogram. In the fourth and fifth cycle, **Tz 1** was exclusively formed, indicating high recycling ability of the nano-template (Fig. 5). It also suggests that the recovered nano-template maintains its templating activity and facilitates the formation of its strong binder from the pool of azide and alkyne fragments.

**Tz 1 selectively binds *c-MYC* G-quadruplex over duplex DNA.** To validate the compound **Tz 1** as the best binding ligand for *c-MYC* G-quadruplex DNA over duplex DNA, we have synthesized the three *in situ* lead products **Tz 1–3** by the Cu(I) catalysed azide-alkyne cycloaddition from their respective alkyne/ azide precursors[38] (Supplementary Figs 13 and 14) and compared their stabilization potential using Förster resonance energy transfer (FRET) based melting assay[39]. We have used 5′-FAM and 3′-TAMRA-labelled *c-MYC* G-quadruplex (100 nM) and a control

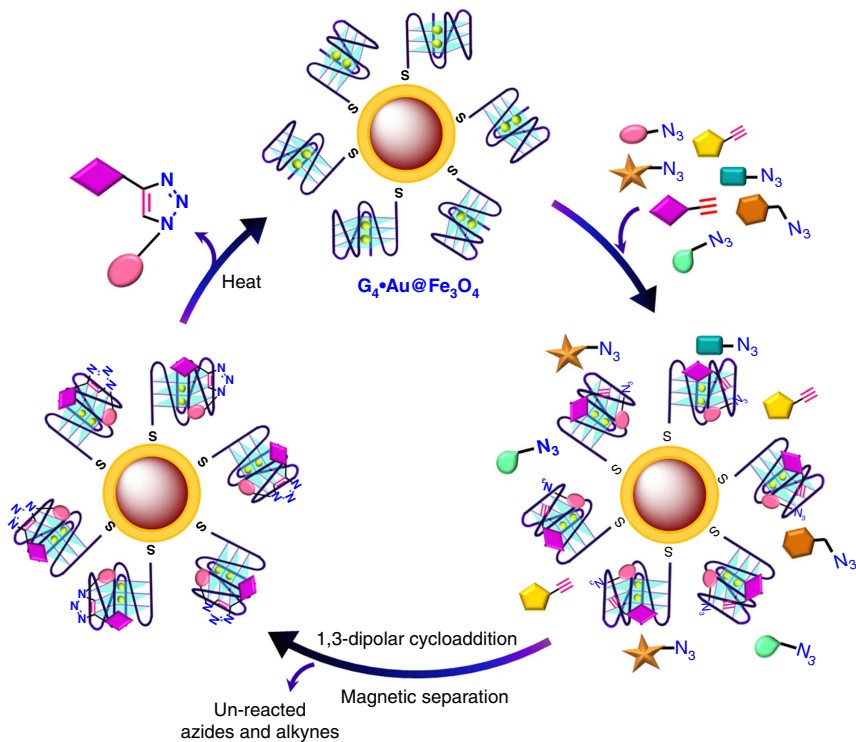

**Figure 5 | Recycling of G₄•Au@Fe₃O₄ nano-template.** Schematic representation of recyclable *c-MYC* G-quadruplex magnetic nano-template catalysed *in situ* screening protocol.

duplex DNA sequence (100 nM) to measure the stabilization potential and selectivity of the triazole products towards the DNA targets. Figure 6a–c demonstrates the melting profile of *c-MYC* and *dsDNA* ($\Delta T_m$) in the presence of various concentration of each compound (Table 1). The compound **Tz 1** showed a maximum stabilization potential for *c-MYC* at only 1 µM ligand concentration ($\Delta T_m = 19.3 \pm 0.96\,°C$) and a negligible stabilization potential towards duplex DNA ($\Delta T_m = 0.86 \pm 0.04\,°C$). The other lead compound **Tz 3** exhibited a $\Delta T_m$ value of $11.14 \pm 0.28\,°C$ for *c-MYC* G₄ DNA and $1.61 \pm 0.12\,°C$ for duplex DNA at 1 µM concentration. Consistent with the duplex DNA templated synthesis, the compound **Tz 2** showed stabilization potential for both *c-MYC* G₄ DNA ($\Delta T_m = 8.17 \pm 0.41\,°C$) and duplex DNA ($\Delta T_m = 8.41 \pm 0.21\,°C$). These observations are in strong agreement with the trend obtained in the DNA•MNP guided synthesis. The FRET melting analysis thus revealed that the compound **Tz 1** showed better stabilization potential ($\Delta T_m$) towards *c-MYC* G-quadruplex DNA over *ds*DNA compared to other triazole derivatives.

FRET competition experiment was performed to evaluate the selectivity of **Tz 1** for the *c-MYC* G-quadruplex DNA over duplex DNA (Supplementary Fig. 15). The $\Delta T_m$ value of ligand **Tz 1** for the *c-MYC* quadruplex showed a negligible change in the presence of 20 equivalent excess of *ds*DNA, indicating high selectivity of **Tz 1** for the quadruplex over duplex DNA. In addition, the circular dichroism (CD) spectra for the *c-MYC* quadruplex show that **Tz 1**, even at high concentrations, did not perturb the parallel conformation of *c-MYC* G-quadruplex DNA (Supplementary Fig. 16).

As the carbazole derivatives are fluorescent, the binding affinities of the *in situ* lead compounds **Tz 1, 2** and **3** for both quadruplex and duplex DNA were investigated by fluorescence titration (Fig. 6d–i). Fluorescence binding titration revealed that these carbazole derivatives are environmentally sensitive probes for the G-quadruplex[36,40]. The major lead compound **Tz 1**

exhibited a selective fluorescence enhancement (3.7-fold) upon binding with G-quadruplex with a 33 nm blueshift, whereas it barely showed any enhancement upon titration with duplex DNA. Compound **Tz 2** showed enhanced fluorescence intensity (2.5-fold) with a significant blueshift of ~40 nm in the presence of both *c-MYC* G-quadruplex and *ds*DNA. The significant blueshift observed for compounds **Tz 1** and **Tz 2** containing the electron donor and acceptor motifs, indicated their binding with the hydrophobic sites of the DNA. In comparison, **Tz 3** produced a slight wavelength change (7 nm redshift) with a 1.9-fold increase in fluorescence intensity upon addition of *c-MYC* G-quadruplex. The fluorescence experiments further indicated that **Tz 1** is selective for the *c-MYC* quadruplex DNA compared to **Tz 2** and **Tz 3** and it has high affinity for *MYC* quadruplex ($K_D = 0.17 \pm 0.01$ µM) (Table 1, Supplementary Fig. 17).

The FRET melting analysis and fluorescence titration results are consistent with the results obtained in *c-MYC* G₄•Au@Fe₃O₄ NP templated target guided synthesis approach. The major lead compound **Tz 1** containing two protonable tertiary amine side chains strongly interacts with quadruplex compared to the second lead compound **Tz 3** that contains one amine side chain. In addition, the minor lead **Tz 2** lacking the aromatic carboxamide group was formed as a major product in the presence of duplex DNA template, which further indicates that compounds with extended aromatic system can selectively target G-quadruplex over duplex DNA. The triazole compound **Tz 1** that showed high affinity for the *c-MYC* G-quadruplex DNA, was selected for carrying out subsequent biological assays to highlight the importance of this methodology to discover important leads for G-quadruplex specific drug development.

**Downregulation of *c-MYC* expression by Tz 1.** We have also assessed the biological activity of **Tz 1** in HCT116 colorectal

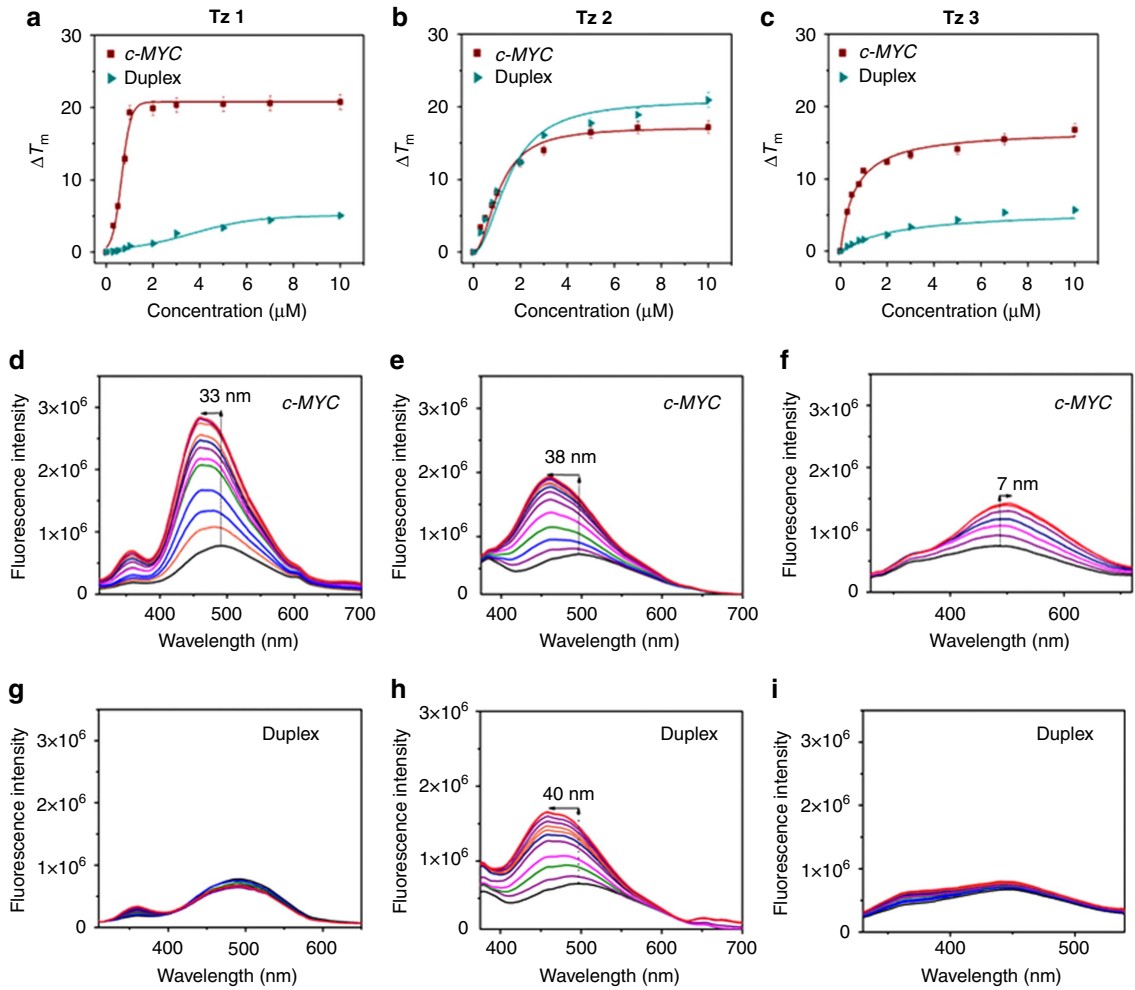

**Figure 6 | Biophysical analyses for the binding of lead compounds to *c-MYC*.** (**a**–**c**) FRET melting profiles of 100 nM *c-MYC* G-quadruplex ($T_m = 70.9 \pm 1.2$ °C) and *ds*DNA ($T_m = 61.3 \pm 2.1$ °C) with increasing amounts of triazole products **Tz 1**, **2** and **3** (0–10 μM) in 60 mM potassium cacodylate buffer, pH = 7.4. (**d**–**i**) Fluorescence responses of **Tz 1**–**3** (1 μM) upon titration with *c-MYC* G-quadruplex DNA and duplex DNA.

**Table 1 | Stabilization potential ($\Delta T_m$) values by FRET melting experiments and dissociation constants measured by fluorescence titration.**

| | $\Delta T_m$(°C) at 1 μM conc. | | | $K_D$(μM) | | |
|---|---|---|---|---|---|---|
| | **Tz 1** | **Tz 2** | **Tz 3** | **Tz 1** | **Tz 2** | **Tz 3** |
| *c-MYC* $G_4$ DNA | $19.3 \pm 0.96$ | $8.17 \pm 0.41$ | $11.14 \pm 0.28$ | $0.17 \pm 0.01$ | $1.13 \pm 0.03$ | $0.51 \pm 0.05$ |
| duplex DNA | $0.86 \pm 0.04$ | $8.41 \pm 0.21$ | $1.61 \pm 0.12$ | $>30$ | $0.88 \pm 0.02$ | $>25$ |

adenocarcinoma cancer cell line. HCT116 has been reported as a model cell line for monitoring the *c-MYC* expression in living cells[41,42]. Fluorescence microscopic images of **Tz 1** treated HCT116 cells reveals that **Tz 1** permeates cell and nuclear membranes and preferentially localizes in the nuclei of the cells (Fig. 7a–c). We then evaluated the short-term growth inhibitory activity of **Tz 1** in HCT116 cell line using MTT assay. Compound **Tz 1** showed strong inhibitory acitivity on the proliferation of HCT116 cells in a dose-dependent manner, with an $IC_{50}$ value of $2.1 \pm 0.1$ μM at 24 h.

Next, we have evaluated the effect of **Tz 1** on the transcriptional and the translational level of *c-MYC* gene. qRT-PCR analysis showed that **Tz 1** reduced the transcription of *c-MYC* gene by 44 and 79% (normalized to GAPDH expression) at 1 and 2 μM concentrations, respectively, compared to the untreated

cells. Western blot analysis corroborated the inhibitory activity of **Tz 1** on the *c-MYC* expression by suppressing the *c-MYC* protein levels by 39 and 74% at 1 and 2 μM treatment, respectively. The expression of GAPDH in HCT116 cells remained unaltered by **Tz 1** (Fig. 7f,g).

To further evaluate whether the observed *c-MYC* inhibition by **Tz 1** was dependent on the ligand mediated G-quadruplex stabilization in the *c-MYC* promoter region, CA46 Burkitt's lymphoma cell line was used to perform the exon-specific assay[41,42]. In CA46 cell line, the transcription of *MYC* exons are differentially regulated; exon 1, which generally remains untranslated, are transcribed under the control of G-quadruplex forming NHE III1 region. Whereas exon 2, which is predominantly responsible for *MYC* protein expression, is independent of G-quadruplex regulation due to the

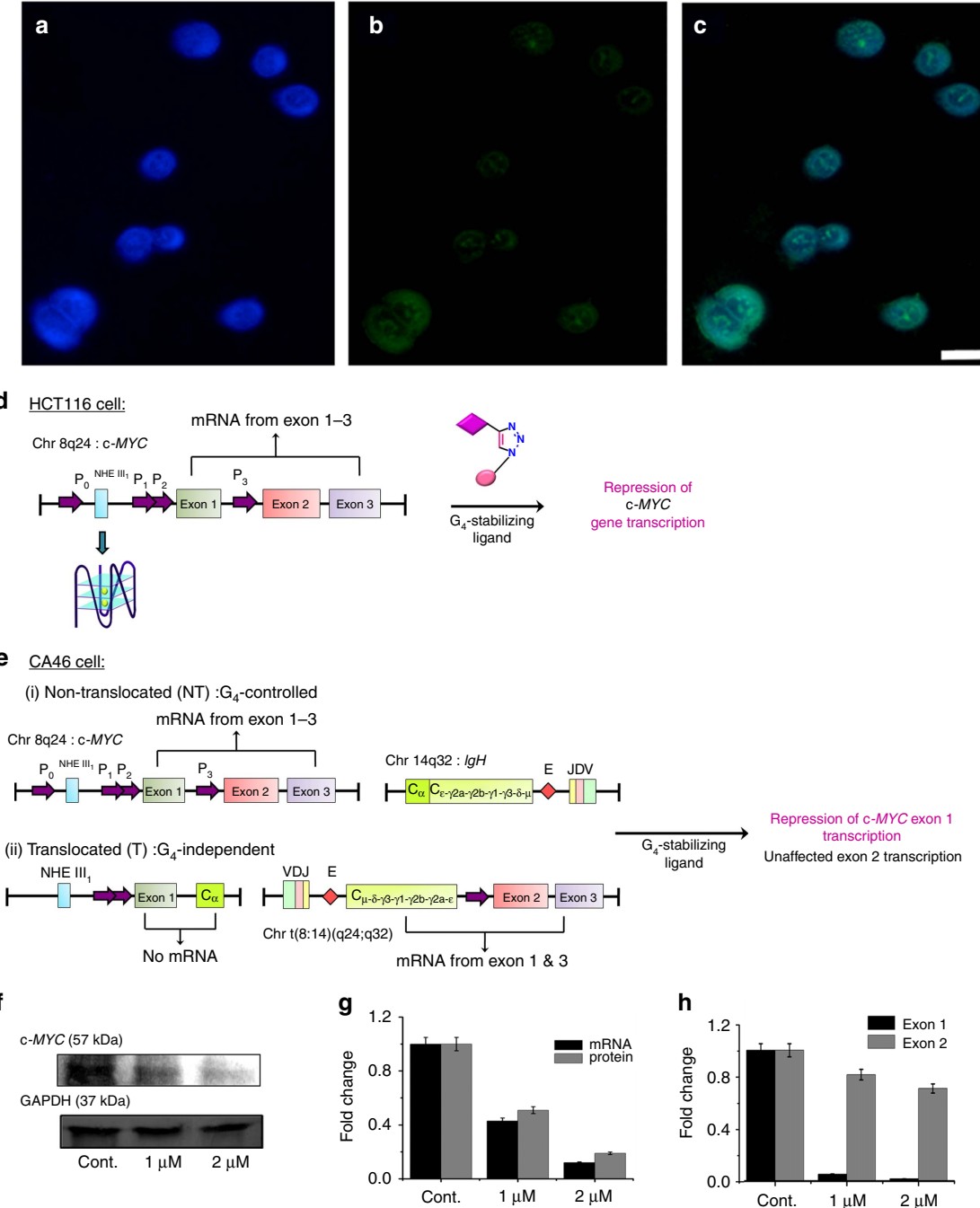

**Figure 7 | Biological validation of Tz 1 as *c-MYC* G$_4$ stabilizer in cancer cells.** Fluorescence microscopic images of HCT116 cells incubated with **Tz 1** and DAPI, Scale bar, 10 μm; cells stained with (**a**) DAPI and (**b**) **Tz 1** and (**c**) co-localization of **Tz 1** and DAPI. (**d**) Schematic representation of G-quadruplex-mediated control over *c-MYC* expression in HCT116 cancer cell line. (**e**) A reciprocal translocation in the CA46 Burkitt's lymphoma cell line between the *IgH* heavy chain gene of chromosome 14 and the *MYC* gene on chromosome 8—NT mRNA expression is under the control of G-quadruplex forming region and T mRNA expression is independent of G-quadruplex. The NHE III$_1$ region moved along with exon 1 produces no mRNA product; therefore represents expression form NT allele; whereas, exon 2 mRNAs are produced from both T and NT allele. (**f**) Immunoreactive bands of *c-MYC* and GAPDH protein expression as analysed by Western blot. (**g**) Concentration dependent downregulation of *c-MYC* mRNA and protein levels in HCT116 cells after treatment with compound **Tz 1** for 24 h. (**h**) Exon-specific qRT-PCR assay with the CA46 Burkitt's lymphoma cell line after 24 h treatment with **Tz 1**. ($n = 3$, ± s.e.m.)

translocation occurred between the chromosome 8 (*MYC* gene) and 14 (IgH heavy chain gene). Thus the transcription of exon 2 mainly represents the *MYC* expression from the G-quadruplex-lost enhancer element of IgH heavy chain. This translocational alteration of the *MYC* gene locus renders the *MYC* protein expression unaffected to the inhibition by G-quadruplex stabilization. Using this model, we have demonstrated the

G-quadruplex stabilizing ability of **Tz 1** by independently monitoring the two *MYC* mRNA products produced within the CA46 cell line upon ligand treatment.

The qRT-PCR assay revealed that **Tz 1** significantly induced exon-specific downregulation of *MYC* expression in a dose-dependent manner. It caused a noteworthy inhibition of exon 1 transcription (95% at 1 μM and 98% at 2 μM

concentration), which is regulated by G-quadruplex formation, while transcription from exon 2 remains unaffected after 24 h treatment with **Tz 1** (Fig. 7h). Thus, on the basis of the observed exon-specific effect of **Tz 1**, we can conclude that **Tz 1** binds and stabilizes the G-quadruplex structure in *c-MYC* P1 promoter and specifically reduces *c-MYC* expression in cancer cells.

It is well documented that *c-MYC* plays a critical role in the regulation of the cellular fate between apoptosis and necrosis and therefore quadruplex interacting molecule that can repress *c-MYC* expression can drive cells towards apoptosis[43]. We investigated the ability of **Tz 1** to induce cell apoptosis in HCT116 cells by flow cytometry. **Tz 1** induced apoptotic cell death in a dose-dependent manner, as demonstrated by the dual staining with propidium iodide (PI) and Annexin V-FITC. The population of the apoptotic cells was increased significantly in the presence **Tz 1**, while there was no remarkable increase of necrotic cells upon 24 h treatment (Supplementary Fig. 18). The observed cellular effect of the lead compound **Tz 1** is likely due to its interaction with the G-quadruplex DNA in the *c-MYC* promoter region.

It may be possible that **Tz 1** can interact with other quadruplexes; among which we have only demonstrated its interaction with *c-MYC* promoter quadruplex DNA. However, this is not surprising as many drugs, even when approved; simultaneously target a set of signalling genes that ultimately lead to apoptosis[44,45]. In this regard, the compound of our interest may exert cellular apoptosis by simultaneous execution of *c-MYC* G-quadruplex stabilization and other secondary cellular effects.

## Discussion

In this work, we have established the use of magnetic nanoparticle-linked *c-MYC* G-quadruplex as a template for covalently assembling potent azide-alkyne partners to generate high-affinity quadruplex ligands. This methodology using DNA nano-templates for metal free Huisgen cycloaddition offers several advantages: First, it reduces the time required for the conventional laborious chemical synthesis of potent candidates and their biological binding/screening assay. Second, it provides a large surface with more number of local DNA catalytic sites for better interaction of the reacting fragments on the nano-surface. Thirdly, it enables efficient isolation of the *in situ* lead compound by simple magnetic decantation and lastly, it prevents the wastage of commercial DNA sequences by allowing the recovery and reuse of DNA magnetic nano-templates. This innovative approach has been used for a model library of azide and alkyne building blocks to obtain two 1,4-triazole lead compounds (out of 33 possible triazole products of 1,4-regiochemistry) for the G-quadruplex target. The major lead compound shows high binding affinity for *c-MYC* G-quadruplex DNA over duplex DNA, localizes exclusively into the nucleus, inhibits the *c-MYC* expression by targeting the *c-MYC* promoter G-quadruplex and induces apoptosis in HCT116 cells. We anticipate that this proof-of-concept study would pave the way for the design and fabrication of biomolecule immobilized magnetic nano-constructs, exhibiting potentially important impacts both in bionanotechnology and biomedical research.

## Methods

**Chemical synthesis.** Detailed procedures for the synthesis of all new compounds and their characterization are provided in Supplementary Methods (For $^1$H and $^{13}$C NMR spectra of all compounds, Supplementary Methods and Supplementary Figs 19–36).

**Synthesis of Fe$_3$O$_4$ nanoparticles.** The Fe$_3$O$_4$ nanoparticles were prepared by co-precipitation technique. This method uses ferric and ferrous ions in a 1:2 molar ratio in basic solutions. An aqueous solution of FeCl$_3$ (4 ml, 1 M) and FeCl$_2$ (1 ml, 2 M in 2 M HCl) was mixed and added into diluted NH$_3$ solution (50 ml, 0.7 M). The reaction mixture was stirred for 30 min at RT. Then, the

precipitate was isolated by magnetic decantation, stirred with diluted HClO$_4$ (50 ml, 2 M) and then collected by centrifugation. The residue was finally made up to 50 ml with deionized water.

**Synthesis and characterization of Au@Fe$_3$O$_4$ nanoparticles.** An aqueous solution of HAuCl$_4$ (2.26 ml, 2.0 mg ml$^{-1}$) was mixed with 15.75 ml deionized water and then heated to boiling. Then, 0.75 ml of Fe$_3$O$_4$ nanoparticle solution was added into the reaction mixture followed by the addition of sodium citrate (0.75 ml, 80 mM) with continuous stirring. The colour of the solution was gradually changed from brown to burgundy. The reaction mixture was heated to 85 °C under stirring for 5 min. After cooling, the solution was sonicated for 15 min, and then Au@Fe$_3$O$_4$ NPs were collected using a magnet, washed three times and redispersed in pure water.

The prepared Au@Fe$_3$O$_4$ NPs were characterized in 100 mM Tris-KCl buffer (pH 7.4) by ultraviolet-visible absorption spectroscopy (Cary Win 300 ultraviolet-visible spectrophotometer using quartz cuvette of path length 1 mm at 25 °C), CD spectroscopy (Jasco J-815), TEM (JEOL 1200 EX electron microscope) and atomic force electron microscopy (NT-MDT instrument in semicontact mode with a resonance frequency of 120 kHz).

**Synthesis of DNA coated Au@Fe$_3$O$_4$ NPs.** All buffers and solutions were bubbled with N$_2$ and degassed to avoid oxidative dimerization of thiolated DNA sequences into disulfide dimers. Thiolated DNA attached nanoparticle probes were synthesized by mixing an aqueous dispersion of Au@Fe$_3$O$_4$ nanoparticles (100 μl) with 100 μl 5′-thiol-capped *c-MYC* G-quadruplex (oligonucleotide concentration is 100 μM in 100 mM Tris-KCl buffer, pH 7.4). After standing for 16 h, the nanoparticles were separated using magnet and washed with 100 mM Tris-KCl buffer (pH 7.4). The volume of the nanoparticle solution was made up to 100 μl with the buffer. Duplex DNA was also attached on the nanoparticle surface by following the same protocol.

The HPLC purified thiol-capped DNAs were supplied by Sigma-Aldrich. The sequences of the DNAs are as follows:
*dsDNA*: [ThiC6]CAAAAATTTTTGCAAAAATTTTTG
*c-MYC*: [ThiC6]GGGGAGGGTGGGGAGGGTGGGG

**β-Mercaptoethanol mediated thiol exchange reaction.** Au@Fe$_3$O$_4$ nanoparticle was surface functionalized with different concentrations of quadruplex DNA (10–50 μM). The DNA attached nanoparticles were then separated by a magnet and washed 3 to 4 times with Tris-KCl buffer, pH 7.4. β-Mercaptoethanol (β-ME) was added (final concentration 12 mM) to the oligonucleotide-modified nanoparticles in 100 mM Tris-KCl buffer (pH 7.4). After 18 h incubation at room temperature with intermittent shaking, the solutions containing the displaced oligonucleotides were separated from the Au@Fe$_3$O$_4$ nanoparticles using an external magnet. The supernatant was then subjected to Native PAGE to determine the surface functionalization of nanoparticles with thiolated oligonucleotides.

**General procedure for G$_4$•Au@Fe$_3$O$_4$ templated cycloaddition.** To a suspension of G-quadruplex nano-template G$_4$•Au@Fe$_3$O$_4$ (10 μl) in 20 μl Tris-KCl buffer (100 mM, pH 7.4), alkynes **1a–c** followed by azides **2–12** were added. The stock solutions of all alkynes and azides were prepared in 5% DMSO solution and then serially diluted in 100 mM Tris-KCl buffer. Final reagent concentrations were as follows: alkyne (0.6 μM), azide (2.4 μM). The mixture containing the nano-template and azide-alkyne building blocks was stirred at room temperature. The cycloaddition products were identified by optimizing the separation protocols. After 2, 4 and 6 days of reaction, the G$_4$•Au@Fe$_3$O$_4$ NPs were separated from the reaction mixture using a magnet and washed thrice with 100 mM Tris-KCl buffer, pH 7.4 (100 μl) to remove the unreacted starting materials. Subsequently, the nano-template was dispersed in 100 mM Tris-KCl buffer, pH 7.4 (50 μl) and the dispersion was then heated for 5 min at 65 °C. The nano-template was again separated instantly by magnetic decantation and the supernatant was analysed by HPLC analysis. The HPLC fractions corresponding to different peaks were identified by ESI-MS spectroscopy.

Control experiment using duplex DNA functionalized nanoparticles (*ds*•Au@Fe$_3$O$_4$ NPs) was performed using similar experimental conditions. The HPLC analysis was carried out using 5.0 μm ODS2 reverse phase column (4.6 × 250 mm) using 254 nm detection wavelength. Flow rate was 0.5 ml min$^{-1}$ CH$_3$CN/H$_2$O (90:10) in 0.1% TFA over 20 min.

**FRET melting analysis.** FRET melting experiments were carried out with a 100 nM oligonucleotide concentration in 60 mM potassium cacodylate buffer, pH 7.4. All HPLC purified dual labelled DNA oligonucleotides were purchased from Sigma-Aldrich. 5′-FAM and 3′-TAMRA-labelled DNA sequences were annealed at a concentration of 200 nM by heating at 95 °C for 5 min followed by gradual cooling to room temperature at a controlled rate of 0.1 °C min$^{-1}$. Dual fluorescently labelled DNA oligonucleotides used in these experiments are:
*c-MYC*: 5′-FAM- GGGGAGGGTGGGGAGGGTGGGG -TAMRA-3′
*dsDNA*: 5′-FAM- CAAAAATTTTTGCAAAAATTTTTG-TAMRA-3′

FRET based screening assay was performed on a real-time PCR apparatus (Light Cycler 480 II System) by incubating the dual labelled DNA oligonucleotides (100 nM) with 1 µM of each triazole derivatives **Tz 1–3** separately for 1 h. Fluorescence measurements were taken with an excitation wavelength of 483 nm and a detection wavelength of 533 nm at intervals of 1 °C over the range of 37–95 °C. Melting temperatures were calculated using Origin Pro 8 data analysis. Titrations experiments were performed by adding different concentration of **Tz 1, Tz 2** and **Tz 3** separately (0.3, 0.5, 0.8, 1.0, 2.0, 3.0, 5.0, 7.0 and 10.0 µM) to 100 nM solution of dual labelled *c-MYC* and duplex DNA in 60 mM potassium cacodylate buffer (pH 7.4). In FRET competition experiment, the concentration of **Tz 1** was kept at 1.0 µM in all sets. The competitor duplex DNA was added to 100 nM *c-MYC* G-quadruplex sequence at final concentration of 100 nM, 500 nM, 1.0 µM and 2.0 µM.

**Circular dichroism spectroscopy.** CD spectra were recorded on a JASCO J-815 spectrophotometer using a 1 mm path-length quartz cuvette. Aliquots of ligand **Tz 1** were added in increments to pre-annealed *c-MYC* G-quadruplex DNA (5′-GGGGAGGGTGGGGAGGGTGGGG-3′) in Tris-KCl (100 mM) buffer at pH 7.4. The DNA concentration used was 10 µM. The CD spectra represented an average of three scans and were smoothed and zero corrected. Final analysis of the data was carried out using Origin 8.0.

**Fluorimetric titration.** The fluorescence spectra were recorded on a Horiba Jobin Yvon Fluoromax 3 instrument at 25 °C in a 10 mm path-length quartz cuvette with filtered 100 mM Tris-KCl buffer (pH 7.4). Fluorescence titrations were performed with successive addition of the DNA solution into the 1 µM ligand solution. The binding constants were calculated using the following Hill 1 equation (1) with Origin Pro 8.0:

$$F = F_0 + \frac{(F_{max} - F_0)[DNA]}{K_D + [DNA]} \quad (1)$$

where $F$ is the fluorescence intensity, $F_{max}$ is the maximum fluorescence intensity, $F_0$ is the fluorescence intensity in the absence of DNA and $K_D$ is the dissociation constant.

In this experiment, following DNA sequences were utilised:
*c-MYC* : 5′-GGGGAGGGTGGGGAGGGTGGGG-3′
*ds*DNA: 5′-CAAAAATTTTTGCAAAAATTTTTG-3′

**Cell culture and viability assay.** HCT116 cell line (ATCC) was a kind gift from Dr Susanta Roychoudhury, Indian Institute of Chemical Biology (Kolkata). CA46 cell line was procured from ATCC. The cells were cultivated at 37 °C in 5% $CO_2$ atmosphere and maintained in DMEM (Dulbecco's modified Eagle Medium, Himedia Cell Culture) + 10% FBS (GIBCO, Heat inactivated, US origin, catalogue number 10082147) and RPMI 1640 (GIBCO, catalogue number 11875093) medium + 20% FBS, respectively, containing 1% anti-anti (GIBCO, catalogue number 15240062). HCT116 cells were passaged by dissociation with 1X trypsin-EDTA (GIBCO, catalogue number 15400054) and CA46 cultures were maintained by centrifugation with subsequent resuspension.

For viability assays, 6,000 cells were seeded per well in 96-well microtiter plates for 24 h. To monitor the ligand induced cell death, HCT116 cells were exposed with various concentrations of triazole **Tz 1** for 24 h. Cell survival was assayed by the MTT method (catalogue number M6494). Absorbance ($OD_{570}$) obtained from control untreated cells was taken as 100% and relative percentage decrease in absorbance was calculated in response to ligand treatment. Data were calculated as the percentage of cell viability. $IC_{50}$ values were determined with Origin Pro 8 software using nonlinear curve fitting.

**Fluorescence microscopy.** The HCT116 cells were plated in six-well plates and treated with 5 µM **Tz 1** for 8 h. After the incubation, the cells were washed with PBS and fixed for 12 min with 1:1 acetone/methanol at −20 °C. The cells were rinsed thrice in PBS, stained with DAPI and kept at room temperature in the dark for 10 min. The cells were then washed twice with PBS, examined and immediately photographed under a fluorescence microscope (Evos FL cell imaging system, ThermoFisher Scientific Ltd.).

**qRT-PCR analysis.** Colorectal cancer cell line HCT116 cells were treated with two different concentrations (1.0 and 2.0 µM) of triazole ligand **Tz 1**. Total RNA from treated and control cells were prepared using the trizol kit according to the manufacturer's protocol (ThermoFisher Scientific, catalogue number 15596018) after 24 h. Reverse transcription was performed using the high capacity cDNA reverse transcription kit (Applied Biosystems, catalogue number 4368814) by following the kit manual. The prepared cDNA was amplified by PCR using Power SYBR Green PCR Master Mix (ThermoFisher Scientific, catalogue number 4367659) with the forward and reverse primers in Light Cycler 480 II (Roche).
Primers used in the amplification of *c-MYC* and GAPDH gene:
*c-MYC* (forward): 5′-CTGCGACGAGGAGGAGGACT-3′
*c-MYC* (reverse): 5′-GGCAGCAGCTCGAATTTCTT-3′
GAPDH (forward): 5′-GACGGCCGCATCTTCTTGT-3′

GAPDH (reverse): 5′-CACACCGACCTTCACCATTTT-3′
The PCR mixture (20 µl) contained 500 nM of each primer, 5 µl of cDNA template and 10 µl of SYBR green mix and the rest of the volume is made up with nuclease free water. The thermal cycling conditions were 10 min at 95 °C and then 40 cycles of 15 s at 95 °C and 60 s at 60 °C. The relative mRNA expression was determined by the arithmetic calibrator ($2^{-\Delta\Delta CT}$) and the difference in *c-MYC* expression was expressed as fold changes.

**Western blot analysis.** After 24 h incubation of HCT116 cells with the above mentioned concentrations of **Tz 1**, cells were washed once with PBS (pH 7.4) and lysed with ice-cold cell lysis buffer (20 mM Tris, 100 mM NaCl, 1 mM EDTA in 0.5% Triton X-100). Cell lysate was collected from the treated and untreated cells, and the total protein content was estimated by Folin–Lowry method. Sixty micrograms of protein from each cell treatment was loaded and separated by 12% SDS–PAGE and transferred to nitrocellulose membrane. The membranes were blocked, washed and probed using 1:700 dilution of anti-*c-MYC* (Invitrogen, catalogue number 710007) and anti-GAPDH antibodies (Invitrogen, catalogue number 398600) for overnight at room temperature. The blots were washed and immunoreactive bands were incubated with a 1:2,000 dilution of respective enzyme conjugated secondary antibodies for 2 h at room temperature. Binding signals were visualized with enzyme substrate (MP) and relative band intensities were determined using ImageJ software.

**Exon-specific assay.** CA46 cells were treated with **Tz 1** at 1 and 2 µM concentration for 24 h. The cells were then washed in PBS twice and the RNA was isolated using the trizol kit according to the manufacturer's protocol (ThermoFisher Scientific, catalogue number 15596018). Reverse transcription of the RNA was performed using the high capacity cDNA reverse transcription kit (Applied Biosystems, catalogue number 4368814). The cDNA was amplified in PCR using Power SYBR Green PCR Master Mix (ThermoFisher Scientific, catalogue number 4367659) with the exon 1 and exon 2 forward and reverse primers in Light Cycler 480 II (Roche) system. For exon 1 and exon 2, the relative mRNA expression was normalized by the GAPDH expression.
Primers:
Exon 1 F: 5′-CACGAAACTTTGCCCATAGC
Exon 1 R: 5′-GCAAGGAGAGCCTTTCAGAG
Exon 2 F: 5′-CCCTCAACGTTAGCTTCACC
Exon 2 R: 5′-AGCAGCTCGAATTTCTTCCA

**FACS analysis for apoptosis.** HCT116 cells were treated with increasing concentrations of **Tz 1** for 24 h and collected by trypsinization. Cell pellet was resuspended in 500 µl 1X Annexin V binding buffer (0.01 M HEPES, pH 7.4, 0.14 M NaCl, 2.5 mM $CaCl_2$) and incubated with 5 µl FITC-Annexin V (ThermoFisher Scientific, catalogue number A13199) and 1 µl 100 µg ml$^{-1}$ propidium iodide (ThermoFisher Scientific, catalogue number P3566) for 15 min. Finally flow cytometry was performed by BD-LSR flow cytometer (BD Biosciences, SanDiego, CA, USA) for apoptosis assay. Approximately 10,000 HCT116 cells were monitored for each sample.

**Data availability.** The authors declare that all data supporting the findings of this study are available in the article and in Supplementary Information file. Additional informations are available from the corresponding author upon request.

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

## Acknowledgements

We dedicate this work to Professor Shankar Balasubramanian on the occasion of his 50th birthday. This research work is supported by the Department of Science and Technology [DST/SJF/CSA-01/2015–16]; and DST Nano Mission [SR/NM/NS-1034/2011(G)], India. J.D. thanks DST for a SwarnaJayanti fellowship; D.P. thanks CSIR India for a research fellowship; T.D. thanks DBT for research fellowship. We gratefully thank Mr Tanmoy Dalui for FACS analysis, IICB Kolkata and Vivek Chander and Dr Arun Bandyopadhyay, IICB for Fluorescence Imaging.

## Author contributions

J.D. and D.P. designed the experiments. D.P. synthesized the compounds and performed HPLC. D.P. and P.S. performed TGS reactions and isolation of the leads. D.P. and P.S. carried out FRET, Fluorescence and CD spectroscopy. P.S. and T.D. performed biological experiments. J.D. supervised the experiments. All authors analysed the data and J.D. and P.S. wrote the manuscript.

## Additional information

**Competing interests:** The authors declare no competing financial interests.

**Publisher's note**: 

