## [Peer Review File · Nature Communications]

Reviewer #1 (Remarks to the Author):

The authors report a recyclable platform for the target-guided synthesis of small molecule biological binders using magnetic nanoparticle-bound DNA. Using this method, the authors detail the discovery, identification, and in vitro validation of a small molecule inhibitor of the c-MYC G-quadruplex sequence, which can be used to induce apoptosis in cancer cells. While this platform has important implications on the field, the manuscript includes overly detailed, procedural descriptions of the experiments and, as a result, it is not of high enough quality for publication in Nature Communications in its current form. Major revisions would be required before it could be resubmitted for reconsideration in Nature Communications.

Specifically:

1. The abstract is very detailed and may benefit from a more concise description of the goals and experimental results.
2. Again, the introduction would benefit from a broader description of the experimental goals (before delving into the experimental detail). Also, the last sentence of the introduction should more directly articulate the hypothesis of the work.
3. Figure 1c,d: These particles do not appear spherical. The size distribution looks very broad. This may not affect the use of these particles as catalysts (as long as their catalytic ability is not being quantitatively measured), but it certainly calls into question whether each of the particles is in fact of the form Au@Fe₃O₄, as opposed to purely Au or Fe₃O₄. Energy dispersive x-ray spectroscopy (EDX) of the final particle products would be useful to obtain a complimentary validation of the structures claimed. This could be resolved by an initial magnetic separation/purification process. Also, how was the size of these particles determined (“The average size of Fe₃O₄ MNPs is 11 nm and after coating with Au, the average particle size is increased to 15 nm”)? If this is done by EM analysis, this should be noted.
4. Although the PAGE experiments may suffer from incomplete particle separation (i.e., any purely Au NPs would remain after magnetic purification), they could be utilized to confirm the surface functionalization. This should be addressed briefly in the main text and all of the experimental details should be transferred to the SI.
5. Figure 3: Schematics can be a concise way of communicating a scientific method or experiment; however, this schematic is overly cartoonistic. It would be helpful if there were a key that delineated the chemical structure symbolized by each shape. This key could also be present in Figure 2 for continuity.
6. Page 7: The paragraph detailing the use of LiCl for lead compound separation should be shortened or transferred to the SI. Why were the results of unsuccessful HPLC experiments included (“lead compounds could not be identified as overlapping peaks were obtained in HPLC analysis”)? It may be worthwhile to explore alternative mobile phases and columns in order to achieve peak

separation. Also, why would one separation process lead to overlapping peaks in HPLC, while a different separation process would result in clearly identifiable peaks? This may suggest that your purification process is affecting the lead compounds.

7. Page 12-14: These control experiments would be better included in the SI. They are very detailed and their results and implications can be summarized in a few sentences in the main text.

8. Conclusion: Although a control particle with double-stranded (as opposed to G quadruplex DNA) provides a valuable control, does the orientation of DNA on the surface of the NPs preferentially catalyze the formation of one lead compound over another? How does the distribution of lead compounds catalyzed by particle-bound DNA compare to free DNA? Is this orientation potentially beneficial (e.g., do the lead compounds catalyzed by the oriented DNA bind more strongly than those produced by free DNA)? While this platform provides an interesting method for reusing catalysts, the manuscript may benefit from a more chemical description of the benefits of the surface-bound DNA.

9. The authors show strong analytical support for most of the claims in the main text of the manuscript. In general, this manuscript would be greatly strengthened by highlighting the key aspects and conclusions of each experiment. In its current form, these higher level ideas are lost in the incredibly detailed text.

Reviewer #2 (Remarks to the Author):

The MS by Panda et al. combines several element for the selection of c-MYC G-quadruplex (MYC-Q) binders.

Gold covered magnetic nanoparticles are synthesized and the MYC-Q are immobilized on the nanoparticles via thiol chemistry. The main purpose of the study is to use the MYC-Q on nanoparticles as the target in Target Guided Synthesis. It is a method where the binding of building blocks from a library to the target templates the formation covalent bonds between building blocks, in this case an alkyne and an azide. In this manner highly selective binders may be formed from libraries of building blocks. A set of building blocks containing 2 alkynes and 7 azides are used in the present study and 3 binders of which 2 are specific are identified. It is verified that the binders have high affinity for MYC-Q and that they down-regulate c-MYC expression and induce apoptosis in cancer cell.

While many aspects of the work are interesting I also have some reservations:

It is claimed in the title and through the MS that the MYC-Q is a catalyst for the for the azide-alkyne reaction of selected substrates. I see no evidence for a turnover >1 and I would say that MYC-Q serves as a template for the reaction rather than a catalyst.

The authors also promote in the title and elsewhere that by immobilizing MYC-Q on the nanoparticles, the MYC-Q is recyclable. However, only one round of recycling is demonstrated and the yields of the products are changed after the first round. Several rounds (>10) should be demonstrated if the recyclability should be of any value.

My most critical reservation is that the library is too small with a total size of only 14 possible combinations. The authors do find good binders, but the fragments are very similar to fragments of compounds that have previously been shown to bind to MYC-Q by the same authors. The identification of the binders, still happens by HPLC and MS and hence it is difficult to see how this should be expanded even to medium-sized libraries of >1000 combinations. The parallel testing of all combinations in individual vials is much easier in spite of the cost of DNA, especially since the product is fluorescent and can immediately be identified. Therefore I have a hard time seeing the potential value of the method.

In conclusion I do not find that the MS is sufficiently important to be published in Nature Communications

Reviewer #3 (Remarks to the Author):

Dash and co-workers report on the development and use of gold nano-particles coated with nucleic acids folded into either double stranded or G-quadruplex DNA structures. These DNA-NPs can be used as catalysts to template the formation of G-quadruplex specific ligands. In particular, the authors are using these catalysts to mediate Huisgen cycloadditions between a library of alkynes and azides with the view to select for the best binder. By doing so, the authors identified a small number of ligands that can stabilize the G-quadruplex motif that can be found in the promoter of MYC oncogene. Then, the authors show that this compound, which is fluorescent, enters cell nuclei and down-regulate MYC at the mRNA and protein levels. The authors show that a short-term treatment induces apoptotic cell death and conclude that this phenotype occurs as a direct consequence of MYC promoter targeting.

The paper is clear and well-written. The authors have used a large diversity of techniques in this study, which is not very common and quite impressive. This include material sciences, supramolecular chemistry, synthetic organic chemistry, confocal microscopy, AFM, TEM, circular dichroism, FRET-melting assays, PCR, western blotting, apoptosis assay. Most of the important controls are also reported. While the work is interesting, I do have some concerns that are listed thereafter.

Major concerns:

1. While it has previously been reported that small molecules can target the G4 in MYC and induce a down-regulation of this gene, there is some scepticism in the field. As yet, no solid proof that the down-regulation of MYC induced by small molecules has anything to do with this structure. There is no doubt that the structure described can form a G4 and can be stabilized by small molecules, as described in this study. However, the concentrations of drugs used to detect significant changes in cells are too high and the effect likely arises as a result of off-target effects. In particular, the authors show by microscopy that the small molecule targets the entire nuclear area. While there are only two copies of the MYC G4 structure, how can the author possibly claim that cell death and MYC down-regulation results from the direct targeting of this structure? The authors are not to be blamed. The entire field publishes such articles and in my opinion, if it is clear that G4 structure in cells have been demonstrated by several labs, in particular by the Balasubramanian laboratory, the existence of isolated, functional/druggable motifs in the promoter region remains controversial and unsubstantiated.

2. The use of NP as solid support to catalyze the Huisgen reaction is novel. However, as stated by the authors, the use of nucleic acids and G4 structures in particular to catalyze this reaction and identify potent binders is not novel. See ref 19 and 20 of this manuscript. It seems that the methodology described here is rather similar to that previously reported by Antonio et al in ref 19. It is not clear what is really novel in this article?

Minor concerns:

1. What is the link between MYC inhibition and induction of apoptosis? Unless cancer cells are addicted to this oncogene, it is not obvious why taking out a transcription factor would initiate cell death. What is the effect of a knock-down of this gene?

2. One advantage of the method appears to be the fact that the catalyst can be recycled. Is this really important? Oligonucleotides are not that expensive. This is a distraction in the paper. The authors should keep the focus on catalysis, turn-over, molecular recognition...

3. Is the probe really a turn on probe or is it just fluorescent? If it is a turn on probe, is it upon binding to G4 or ds DNA? What are the consequences on the interpretation of the confocal microscopy data?

4. In relation to my previous point, the authors claim that the effect is linked to the targeting of a single G4 motif while the targeting appears to be pan-nuclear. Another, and in my opinion more relevant, angle would be that this molecule is a pan-G4 binder, which would explain why the staining is pan-nuclear. There is no way this compound can be selective to the G4 in MYC and not target the other 750.000 identified by sequencing (see Nat Biotech, Balasubramanian et al cited in this paper). In particular, this compound could well target all these motifs and induce DNA damage in a replication/transcription-dependent manner. This would explain the apoptosis reported by the authors. This has previously been described by few groups including the Jackson lab (Nat Chem Biol 2012). The authors should look at some DNA damage marks after a short period treatment and see whether the DDR is activated. This should be done at a time point that precedes apoptosis, of course.

5. The authors are highlighting the value of their method, which is very fair. However, TGS with DNA has been reported. The authors here claim that it takes 6 days to select for small molecules using G4-NP in absence of copper I assume. The ref 19 describes similar reaction times. Thus, the NP strategy does not appear to have a much higher turn-over. What happens with lower amounts of NP and shorter time period? To the very least, the authors should compare the methods experimentally to make such a claim. This could, in fine, demonstrate the advantage of NP vs solution TGS with nucleic acids.

6. Is there any selection with a mutated MYC sequence that cannot form a G4?

7. What is the value of using TEM and AFM data? Is it really important to have homogeneous mono-disperse material for an effective catalysis (which is the main point of this paper if I understand correctly)?

8. The CD spectra in SI are not normalized on the y-axis. CD/mdeg is not a unit and does not take into account the experimental settings including concentration of the oligos.

9. Figure S2. The gel contains no control oligo ladders. The control used shows several bands? Also, the quality of the gel presented is not great and could be improved.

10. Did the authors attempt to add a copper catalyst in addition to the G4-NP? While the NP could promote the correct selection, one could expect that copper reduces the time of the reaction (perhaps at the expense of the selection).

11. In terms of molecular recognition, it is indeed conceivable that the hetero-aromatic carbazole interacts by means of pi-pi interactions but the surface covered by this fragment is rather small and could equally interact with a single DNA base. Thus, the ds-DNA could in principle also select for good binders. Is this happening? If not, what is the rationale behind this. If it does, are the selected compounds interacting with genomic DNA in live cells and what is the effect?

12. Same comment for electrostatic interactions. The phosphate backbone can be found also in ds-DNA. Why should these molecules selectively interact with the G4?

13. While it might be obvious, the authors should explain their choices of fragments employed. Since the library is rather small (it does not need to be large), it may be wise to comment on types of supramolecular interactions these functions can engage rather than the functional groups themselves.

14. Heating at 65°C (even with the presence of LiCl) may not be stringent enough to release the binders from the NP, especially if the ligands identified are tight binders. This can introduce biases towards the analysis, or what appears to be the likely outcome of the selection process. How can one be sure that the compounds have been quantitatively recovered from the NPs?

15. The authors mention that Li prevent recycling of the NP. Can the G4 not be re-folded in the presence of potassium? If not, what are the reasons?

16. One could use, as previously described, a biotinylated oligo, and recover the ligands similarly. It would be more simple. What do gold/thiol provide over biotin/streptavidin?

17. What happens when the fragments leading to the lead compounds are removed from the selection process? Did the authors detect other, less efficient, binders? If so, are these still biologically active?

18. Line 262: 'traizole' should read 'triazole'.

19. The melting temperatures observed are not particularly elevated. I do not think it matters so much whether it is 8 or 30 K. I would perhaps moderate the wording.

20. What is the rationale for the 'high' melting temperature induced by compound 21? The compound was not selected by in situ click with G4 if I understand correctly.

21. Also, why the melting temperature induced by compound 10 is similar for both ds and G4 DNA while it was not obtained with the G4 in situ? What is the effect of this compound on MYC?

22. The scale bar in Fig 7a is missing. The quality of DAPI is not great. A data without DAPI would be welcome. As it is, we do not know what the effect of DAPI is on the nuclear staining of the compound alone. As mentioned above, this data shows a pan-nuclear staining that looks like the ds minor groove binder DAPI. This suggests that the compound interacts with many more DNA genes/G4/ds than the paper would have one to believe. Thus, the question is whether MYC is down-regulated as a result of direct targeting or something else, for example a global DNA damage induction that would also explain apoptosis.

23. Annexin/PI should be backed up by PARP and Caspase 3 cleavage, biochemical markers of the cell death pathway. Also, the inhibitor ZVAD should rescue some level of viability in cells treated with the ligand.

Responses to the Reviewers:

Reviewer #1 :

The authors report a recyclable platform for the target-guided synthesis of small molecule biological binders using magnetic nanoparticle-bound DNA. Using this method, the authors detail the discovery, identification, and in vitro validation of a small molecule inhibitor of the c-MYC G-quadruplex sequence, which can be used to induce apoptosis in cancer cells. While this platform has important implications on the field, the manuscript includes overly detailed, procedural descriptions of the experiments and, as a result, it is not of high enough quality for publication in Nature Communications in its current form. Major revisions would be required before it could be resubmitted for reconsideration in Nature Communications.

Specifically:

1. The abstract is very detailed and may benefit from a more concise description of the goals and experimental results.

-As per the requirement, we have concised and modified the abstract and the manuscript accordingly.

2. Again, the introduction would benefit from a broader description of the experimental goals (before delving into the experimental detail). Also, the last sentence of the introduction should more directly articulate the hypothesis of the work.

-We have modified the introduction.

3. Figure 1c,d: These particles do not appear spherical. The size distribution looks very broad. This may not affect the use of these particles as catalysts (as long as their catalytic ability is not being quantitatively measured), but it certainly calls into question whether each of the particles is in fact of the form Au@Fe₃O₄, as opposed to purely Au or Fe₃O₄. Energy dispersive x-ray spectroscopy (EDX) of the final particle products would be useful to obtain a complimentary validation of the structures claimed. This could be resolved by an initial magnetic separation/purification process. Also, how was the size of these particles determined ("The average size of Fe₃O₄ MNPs is 11 nm and after coating with Au, the average particle size is increased to 15 nm")? If this is done by EM analysis, this should be noted.

-After the preparation of the Au coated Fe₃O₄ nanoparticles, the NPs were extensively washed with deionized water to remove the free water soluble gold particles. The Au@Fe₃O₄ NPs were then isolated by magnetic decantation and subjected to TEM analysis. To validate the compositional entities of these NPs, we have performed EDX of each nanoparticle and found both Au and Fe in the NP spectra, as provided in the supporting figure S1b. The particles look nearly spherical.

The average diameter of the NPs was determined by TEM software Gatan Digital Micrograph and AFM software Image Processing and Analysis 3.5.0.2060 program. The instrumentation details are now included in the modified manuscript.

4. Although the PAGE experiments may suffer from incomplete particle separation (i.e., any purely Au NPs would remain after magnetic purification), they could be utilized to confirm the surface functionalization. This should be addressed briefly in the main text and all of the experimental details should be transferred to the SI.

-Before the attachment of DNA, we have extensively washed the Au coated Fe NPs to remove the water soluble pure Au NPs as described in the SI. Thus there is least chance of surface functionalization of pure Au NPs as it is already removed.

As per the suggestion, the experimental details has been included in the revised S.I.

5. Figure 3: Schematics can be a concise way of communicating a scientific method or experiment; however, this schematic is overly cartoonistic. It would be helpful if there were a key that delineated the chemical structure symbolized by each shape. This key could also be present in Figure 2 for continuity.

-We have incorporated some of the symbolized structures in our schematic representation since all the 9 sructures are difficult to incorporate.

6. Page 7: The paragraph detailing the use of LiCl for lead compound separation should be shortened or transferred to the SI. Why were the results of unsuccessful HPLC experiments included (“lead compounds could not be identified as overlapping peaks were obtained in HPLC analysis”)? It may be worthwhile to explore alternative mobile phases and columns in order to achieve peak separation. Also, why would one separation process lead to overlapping peaks in HPLC, while a different separation process would result in clearly identifiable peaks? This may suggest that your purification process is affecting the lead compounds.

-As per the suggestions, the comparative HPLC experiments of the separation process are transferred to the S.I. In the first separation process, the reaction supernatant contains all the azide alkyne building blocks along with the newly generated product. In this process, only the DNA/NP construct have been removed from mixture after heating and therefore overlapping peaks of all the compounds appeared in HPLC. In comparison, only the DNA/NP bound newly generated products have been isolated by magnetic purification in separation protocol 2 (described in Fig. 3). Therefore, clearly separated peaks have been identified.

We have used reverse phase column and water/acetonitrile as mobile phase because the template synthesis with DNA/NP system has been carried out in aqueous solution. As it is desirable and mandatory that the drug component should be soluble in water we have specifically utilised the above-mentioned mobile phase and column to identify the water soluble lead compounds.

7. Page 12-14: These control experiments would be better included in the SI. They are very detailed and their results and implications can be summarized in a few sentences in the main text.

-The results have been briefed in the manuscript and included in the S.I.

8. Conclusion: Although a control particle with double-stranded (as opposed to G quadruplex DNA) provides a valuable control, does the orientation of DNA on the surface of the NPs preferentially catalyze the formation of one lead compound over another? How does the distribution of lead compounds catalyzed by particle-bound DNA compare to free DNA? Is this orientation potentially beneficial (e.g., do the lead compounds catalyzed by the oriented DNA bind more strongly than those produced by free DNA)? While this platform provides an interesting method for reusing catalysts, the manuscript may benefit from a more chemical description of the benefits of the surface-bound DNA.

We have examined our building blocks with free G4 DNA. As we have mentioned in the article, similar lead compounds 13 and 14 were obtained with free G4 DNA templated synthesis. However, as expected, we have observed overlapping spectra for the products along with unreacted azide-alkyne fragments; so we cannot measure the exact distribution of the lead compounds.

As per our knowledge, the orientation of the NP bound DNA and the free DNA did not have any effect on the formation of lead compound (evidenced by CD spectroscopy Fig 1b). Here, the DNA was immobilized on the nanoparticle support for the purpose of recycling the same DNA molecule. The DNA did not change its folded conformation upon attachment on the NP surface as indicated by CD spectroscopy. Therefore, both of them would generate similar product distribution. Also it is important to mention, that DNA immobilization on NP surface provides a large surface area with numbers of localized nano-template centres that would speed up the reaction rate and produce a higher yield.

9. The authors show strong analytical support for most of the claims in the main text of the manuscript. In general, this manuscript would be greatly strengthened by highlighting the key aspects and conclusions of each experiment. In its current form, these higher level ideas are lost in the incredibly detailed text.

-We have modified the manuscript by highlighting our main focus in this paper.

Reviewer #2 (Remarks to the Author):

The MS by Panda et al. combines several element for the selection of c-MYC G-quadruplex (MYC-Q) binders. Gold covered magnetic nanoparticles are synthesized and the MYC-Q are immobilized on the nanoparticles via thiol chemistry. The main purpose of the study is to use the MYC-Q on nanoparticles as the target in Target Guided Synthesis. It is a method where the binding of building blocks from a library to the target templates the formation covalent bonds between building blocks, in this case an alkyne and an azide. In this manner highly selective binders may be formed from libraries of building blocks. A set of building blocks containing 2 alkynes and 7 azides are used in the present study and 3 binders of which 2 are specific are identified. It is verified that the binders have high affinity for MYC-Q and that they down-regulate c-MYC expression and induce apoptosis in cancer cell.

While many aspects of the work are interesting I also have some reservations:

It is claimed in the title and through the MS that the MYC-Q is a catalyst for the for the azide-alkyne reaction of selected substrates. I see no evidence for a turnover >1 and I would say that MYC-Q serves as a template for the reaction rather than a catalyst.

-We have mentioned our system as template rather than catalyst in our revised manuscript.

The authors also promote in the title and elsewhere that by immobilizing MYC-Q on the nanoparticles, the MYC-Q is recyclable. However, only one round of recycling is demonstrated and the yields of the products are changed after the first round. Several rounds (>10) should be demonstrated if the recyclability should be of any value.

-We have examined the recyclability of the system for 5 times and the yields are more-or-less similar (data not shown in the manuscript). However, it is noteworthy to mention that the main lead compound showed the highest yield in every recycling process.

My most critical reservation is that the library is too small with a total size of only 14 possible combinations. The authors do find good binders, but the fragments are very similar to fragments of compounds that have previously been shown to bind to MYC-Q by the same authors. The identification of the binders, still happens by HPLC and MS and hence it is difficult to see how this should be expanded even to medium-sized libraries of >1000 combinations. The parallel testing of all combinations in individual vials is much easier in spite of the cost of DNA, especially since the product is fluorescent and can immediately be identified. Therefore I have a hard time seeing the potential value of the method.

-Inspired by the previous papers of G-quadruplex binding ligands, we have used representative azides and also included new non-polar fragments so that we could obtain all possible types of triazole products (28 compounds including 1,4 (14 products) and 1,5 (14 products) regioisomers that can selectively bind to the target.

Since it is a target guided synthesis approach, it will definitely choose its specific binders from the pool of reaction fragments, independent of the size of the library.

First of all, synthesis of a huge library of potential compounds and testing their individual activity is very time-consuming and laborious. Secondly, this individual screening procedure should not depend on a single experiment rather than different biophysical methods and it consumes a huge amount of DNA during the experiments.

The advantage of utilising the recyclable DNA templates in our procedure over the normal screening is to combine both the synthesis and screening procedure in one step and consequently it reduces the consumable expenses and time. We firmly believe that this method is able to find the specific ligand for a particular target in a better way than the conventional methods.

In conclusion I do not find that the MS is sufficiently important to be published in Nature Communications.

It is easy to identify and isolate the products by using NP-supported TGS compared to conventional solution TGS. This work also delineates the first example of a NP-immobilized DNA for templating TGS for the identification of the best binder from azide and alkyne fragments leading to regioselective formation of the triazole ligands. The use of NP as solid support to template the Huisgen reaction is novel.

Reviewer #3 (Remarks to the Author):

Dash and co-workers report on the development and use of gold nano-particles coated with nucleic acids folded into either double stranded or G-quadruplex DNA structures. These DNA-NPs can be used as catalysts to template the formation of G-quadruplex specific ligands. In particular, the authors are using these catalysts to mediate Huisgen cycloadditions between a library of alkynes and azides with the view to select for the best binder. By doing so, the authors identified a small number of ligands that can stabilize the G-quadruplex motif that can be found in the promoter of MYC oncogene. Then, the authors show that this compound, which is fluorescent, enters cell nuclei and down-regulate MYC at the mRNA and protein levels. The authors show that a short-term treatment induces apoptotic cell death and conclude that this phenotype occurs as a direct consequence of MYC promoter targeting.

The paper is clear and well-written. The authors have used a large diversity of techniques in this study, which is not very common and quite impressive. This include material sciences, supramolecular chemistry, synthetic

organic chemistry, confocal microscopy, AFM, TEM, circular dichroism, FRET-melting assays, PCR, western blotting, apoptosis assay. Most of the important controls are also reported. While the work is interesting, I do have some concerns that are listed thereafter.

Major concerns:

1. While it has previously been reported that small molecules can target the G4 in MYC and induce a down-regulation of this gene, there is some scepticism in the field. As yet, no solid proof that the down-regulation of MYC induced by small molecules has anything to do with this structure. There is no doubt that the structure described can form a G4 and can be stabilized by small molecules, as described in this study. However, the concentrations of drugs used to detect significant changes in cells are too high and the effect likely arises as a result of off-target effects. In particular, the authors show by microscopy that the small molecule targets the entire nuclear area. While there are only two copies of the MYC G4 structure, how can the author possibly claim that cell death and MYC down-regulation results from the direct targeting of this structure? The authors are not to be blamed. The entire field publishes such articles and in my opinion, if it is clear that G4 structure in cells have been demonstrated by several labs, in particular by the Balasubramanian laboratory, the existence of isolated, functional/druggable motifs in the promoter region remains controversial and unsubstantiated.

-We agree with the reviewer that the structure stabilization may or may not lead to the down regulation. But we have given subsequent experimental proofs (Western blot, mRNA) along with the biophysical data where the fluorescence, FRET and CD data shows the stabilization of c-MYC quadruplex DNA by compound 14 followed by the biological experiments that established the down regulation of c-MYC gene.

Further to validate the specific gene downregulation, we have performed the CA46 exon specific assay (*J. Med. Chem.* **2012**, *55*, 6076–6086) that conclusively indicates the downregulation of c-Myc gene by compound 14 (unpublished results, data not shown in the manuscript)

In recently published papers, the G-quaruplex binding ligands show the IC₅₀ value ranging from 0.5-10 μM (*J. Med. Chem.* 2011, *54*, 5671–5679, *ACS Chem. Biol.*, **2016**, *11* (1), pp 139–148). In our paper, the IC₅₀ value of adduct 14 in HCT116 cells (2.1 μM) is comparable with those peer reviewed publications. And thus we have performed the other experimental assays below the apoptotic dose (< 2 μM).

The Adduct 14 itself is highly fluorescent molecule. All the molecules enter to the nucleus and transmit their own fluorescence upon excitation and thus the entire nucleus turns green. In our opinion it is very difficult to distinguish the fluorescence difference between the G quadruplex binding adduct and the free adduct molecules by fluorescence microscopy. This experiment only ensures the permeability of the ligand through the nuclear membrane which is a basic necessity of a DNA binding druggable compound.

2. The use of NP as solid support to catalyze the Huisgen reaction is novel. However, as stated by the authors, the use of nucleic acids and G4 structures in particular to catalyze this reaction and identify potent binders is not novel. See ref 19 and 20 of this manuscript. It seems that the methodology described here is rather similar to that previously reported by Antonio et al in ref 19. It is not clear what is really novel in this article?

-The solution phase *in situ* click reaction with G-quadruplex DNA has been reported by Antonio et al and Hu et al. Our study provides several advantages over the published methods by using this DNA/NP system:

First, (i) we have prepared and characterized magnetic nano supported G-quadruplex and duplex templates that provides a larger surface area with more numbers of localized reaction centres that would speed up the rate of reaction between small molecule fragments with complementary reacting groups.

Secondly, the *in situ* lead compounds can be efficiently isolated by simple heating followed by magnetic decantation. Moreover, the system enables the recycling of the same template DNA molecule and consequently reduces time and cost. This work therefore delineates the first example of a nanoparticle supported TGS for the identification of potent regioselective triazole ligands over the existing solution TGS with nucleic acid.

We believe that this recyclable nano-template approach can also be applicable with other biomolecules other than G-quadruplex DNA to easily identify the target specific binders.

Minor concerns:

1. What is the link between MYC inhibition and induction of apoptosis? Unless cancer cells are addicted to this oncogene, it is not obvious why taking out a transcription factor would initiate cell death. What is the effect of a knock-down of this gene?

As per the literature c-Myc gene is over expressed in HCT 116 cells (*J. Med. Chem.* 2012, *55*, 6076–6086) and this is one of the hallmark of cancer (*Oncogene* (2008) **27**, 6462–6472). There are evidences where inhibition of cMYC expression induces apoptosis (*Mol. Cell. Biol.*, 1999, *19*, 1-11, *Oncogene* (1999) *18*, 2967- 2987).

Over expression of a major transcription factor may lead to the abnormal cell growth as well as inhibition of the expression may rescue the normal growth to some extent. Knockdown of c-MYC may lead to the inhibition of the cell growth and gradually may lead to the cell death (Breast Cancer Res. 2005; 7(2): R220–R228). Herein we have demonstrated that our ligand of interest bind to the cMYC G-quadruplex and reduces the expression of cMYC gene. Further this ligand induces apoptosis in HCT116 cells. However the exact molecular mechanism is under investigation.

2. One advantage of the method appears to be the fact that the catalyst can be recycled. Is this really important? Oligonucleotides are not that expensive. This is a distraction in the paper. The authors should keep the focus on catalysis, turn-over, molecular recognition...

-We would like to highlight the point of target guided synthesis with biologically relevant target based on molecular recognition of reactive azide-alkyne fragments. However, in this methodology we have included a nanoparticle support that promotes the recycling of the DNA-nanotemplate. In this context, we have mentioned the recycling ability of the system that reduces the cost of commercial DNAs that are required in high amount to perform the conventional screening techniques.

3. Is the probe really a turn on probe or is it just fluorescent? If it is a turn on probe, is it upon binding to G4 or ds DNA? What are the consequences on the interpretation of the confocal microscopy data?

Free Adduct 14 is itself a fluorescent molecule. In our experiment we found that it shows 5 fold increases in fluorescence intensity with cMyc G4 DNA (Figure 6g). Compared to that we did not found significant increase in fluorescence when it is targeted to the ds DNA(Figure 6h). In that respect we definitely mark Adduct 14 as a “turn on” probe in vitro.

The cellular system is much more complex and diverse compared to the in vitro system. Binding cannot be firmly concluded only from the microscopic image. Here our goal is to show that the ligand permeabilizes the nuclear membrane and localizes into the nucleus after entering into the cell which is an obvious criteria for a DNA binding drug.

4. In relation to my previous point, the authors claim that the effect is linked to the targeting of a single G4 motif while the targeting appears to be pan-nuclear. Another, and in my opinion more relevant, angle would be that this molecule is a pan-G4 binder, which would explain why the staining is pan-nuclear. There is no way this compound can be selective to the G4 in MYC and not target the other 750.000 identified by sequencing (see Nat Biotech, Balasubramanian et al cited in this paper). In particular, this compound could well target all these motifs and induce DNA damage in a replication/transcription-dependent manner. This would explain the apoptosis reported by the authors. This has previously been described by few groups including the Jackson lab (Nat Chem Biol 2012). The authors should look at some DNA damage marks after a short period treatment and see whether the DDR is activated. This should be done at a time point that precedes apoptosis, of course.

-In this paper we are particularly interested to evaluated the binding selectivity of the target guided synthesized compound towards quadruplex DNA over duplex DNA. Here cMyc just represents one type of Quadruplex DNA.

5. The authors are highlighting the value of their method, which is very fair. However, TGS with DNA has been reported. The authors here claim that it takes 6 days to select for small molecules using G4-NP in absence of copper I assume. The ref 19 describes similar reaction times. Thus, the NP strategy does not appear to have a much higher turn-over. What happen with lower amounts of NP and shorter time period? To the very least, the authors should compare the methods experimentally to make such a claim. This could, in fine, demonstrate the advantage of NP vs solution TGS with nucleic acids.

-Generally 24 hr is required to complete a Cu catalysed Huisgen cycloaddition reaction. Various groups working in the TGS field have performed the in situ reaction for 6 days at RT (J. AM. CHEM. SOC. 9 VOL. 126, NO. 40, 2004). So we followed similar protocol to select small molecules using G4-NP. Yes, it is true; we have not checked the effect of lower amount of NP and shorter time period. Since we intended to obtain a new screening procedure and we do not have any previous knowledge about the lead compound, in this case we have taken 6 days incubation to allow all the possible combinations of azide-alkyne fragments and interestingly our modified method have picked the best lead compound among the all possible combinations.

6. Is there any selection with a mutated MYC sequence that cannot form a G4?

-In the manuscript, we have used a cMYC quadruplex nano template and a duplex nano template, for identifying selective ligands for quadruplexes over the duplex DNA.

We have also explored the binding ability of this ligand in a single stranded non-quadruplex DNA (apart from dsDNA) and not observed any significant changes in fluorescence titration (results were not included in the manuscript).

7. What is the value of using TEM and AFM data? Is it really important to have homogeneous mono-disperse material for an effective catalysis (which is the main point of this paper if I understand correctly)?

-The homogenous mono-disperse material is not required for target guided synthesis. We have shown the data for the characterization of the NPs.

8. The CD spectra in SI are not normalized on the y-axis. CD/mdeg is not a unit and does not take into account the experimental settings including concentration of the oligos.

-We have followed the experimental methods as per the mentioned publications. (*Chem. Commun.*, 2012, **48**, 7607-7609, *Nucleic Acids Res.* 2012 Nov 1;40(20):10334-44)

9. Figure S2. The gel contains no control oligo ladders. The control used shows several bands? Also, the quality of the gel presented is not great and could be improved.

-The control lane showed several bands in this 20% Native PAGE which may correspond to the different folded conformations of c-MYC G-4 DNA.

10. Did the authors attempt to add a copper catalyst in addition to the G4-NP? While the NP could promote the correct selection, one could expect that copper reduces the time of the reaction (perhaps at the expense of the selection).

-Our main goal is to identify the best ligand by G4-NP targeted synthesis. Theoretically addition of copper speeds up the reaction but will generate all the possible 1,4-triazoles and the true ligand may not be distinguished. So, this will not serve our purpose.

Moreover addition of copper only produces the 1,4-regioisomer. We definitely intended to know the type of the regio-isomerism of the resulting triazole ligand. Therefore, we have not checked the reaction with copper. Now once we have found that our resulting ligand belongs to the 1,4 regioisomer class and we did not get any 1,5-product. Therefore we can now easily synthesize the lead ligand in large scale by copper catalysed cycloaddition reaction for biological evaluation purposes.

11. In terms of molecular recognition, it is indeed conceivable that the hetero-aromatic carbazole interacts by means of pi-pi interactions but the surface covered by this fragment is rather small and could equally interact with a single DNA base. Thus, the ds-DNA could in principle also select for good binders. Is this happening? If not, what is the rationale behind this. If it does, are the selected compounds interacting with genomic DNA in live cells and what is the effect?

-The individual alkyne and azide fragments can interact with G4-DNA and dsDNA. In TGS, only those fragments come in proper orientation on binding to G4 or ds DNA that lead to the formation of potent target binding ligands. In our case we have obtained 3 leads for G4-DNA and only one of them has been identified for dsDNA. Here, the azide 3, 6 and 7 are individually able to interact with the DNA phosphate backbone. But when they form the triazole product, their binding affinity has been changed due to their gained aromaticity. Since lead 13 and 14 contain conjugated aromatic rings, which diminishes its interaction with dsDNA and further aids in the stacking interaction with the planar quadruplex. In comparison ds DNA can only generate triazole 10 due to the lack of para substituted benzene ring. In the three lead compounds, the planar heteroaromatic carbazole core can undergo stacking interactions with the planar bases. It is the components of the azide fragments that determine their binding selectivity.

12. Same comment for electrostatic interactions. The phosphate backbone can be found also in ds-DNA. Why should these molecules selectively interact with the G4?

-We have obtained compound 10 by dsDNA targeted TGS. As we have mentioned above, here the components of the azide fragments play key role in determining the selectivity but only when they are conjugated in the triazole product. The amine group of azides 3, 6 and 7 is responsible for the electrostatic interaction with the phosphate backbone. So azide 3 derived compound 10 non-selectively binds both G4-DNA and dsDNA. But the conjugated aromatic ring of azide 6 and 7 further aided in stacking interaction with the external G-quartet and also prevented the interaction with dsDNA (compound 13 and 14).

13. While it might be obvious, the authors should explain their choices of fragments employed. Since the library is rather small (it does not need to be large), it may be wise to comment on types of supramolecular interactions these functions can engage rather than the functional groups themselves.

-We have included these points in the revised manuscript. The functions of these fragments are as follows
Az 2 and 3: containing aliphatic amine to interact with the phosphate backbone. Az 3 will interact with phosphate backbone better than azide 2 due to its protonation of NMe₂ group by inductive effect.
Az 4: aid in π - π stacking interactions with the G-quartet due to the aromatic ring

Az 5: help in π - π stacking interactions (the aromatic ring) and weak electrostatic interactions (Boc protected prolinamide).

Az-6: aid in π - π stacking interactions with the G-quartet due to the aromatic ring and the $-NH_2$ group for electrostatic interactions

Az-7: aromatic carboxamide with amine side chains for π - π stacking and electrostatic interactions

Az-8: Guanosine groups can aid to stack upon the G-quadruplex bases.

14. Heating at 65°C (even with the presence of LiCl) may not be stringent enough to release the binders from the NP, especially if the ligands identified are tight binders. This can introduce biases towards the analysis, or what appears to be the likely outcome of the selection process. How can one be sure that the compounds have been quantitatively recovered from the NPs?

-We cannot use high temperature because of the detachment of the thiolated DNA molecules from the nano-surface at higher temp. (*Colloids Surf B Biointerfaces*. 2006 Aug 15;51(2):130-9). Thus we cannot reuse the system if we apply temperature higher than 65°C.

Alternatively, we repetitively performed the separation protocol 2 until obtained full recovery, confirmed by HPLC.

15. The authors mention that Li prevent recycling of the NP. Can the G4 not be re-folded in the presence of potassium? If not, what are the reasons?

-Li⁺ destabilises the G4 conformation as it cannot stabilize the Hoogsteen H-bonding (*Nucleic Acids Res* 32, 2598-2606 (2004)). In this case we used 8M LiCl which is too high compared to the K⁺ present in the (100 mM Tris KCl solution). Thus G4 conformation will remain unfolded

16. One could use, as previously described, a biotinylated oligo, and recover the ligands similarly. It would be more simple. What do gold/thiol provide over biotin/streptavidin?

-The use of biotinylated oligo (expensive) is also simple, but our system is quite new and advantageous because of the following reason.

The system, as we have mentioned in the manuscript, provides a number of recyclable reaction centres on a single surface that would speed up the TGS which is not possible with the biotinylation.

17. What happens when the fragments leading to the lead compounds are removed from the selection process? Did the authors detect other, less efficient, binders? If so, are these still biologically active?

-As per the figure 2, the combination of alkyne 1a and azide 3, 6 and 7 gives the lead compounds (Fig. 4) amongst which the triazole product of alkyne 1a and azide 7 has been generated in higher yield. Since we did not obtain any triazoles derived from alkyne 1b, thus we may confer that in the absence of alkyne 1a, there will be no significant product. On the other hand, if we remove the above-mentioned azides from the reaction, azide 2 and 8 may bind and take part in the TGS as per the previously reported literatures. However, their binding efficiencies and functionalities are expected to be low as per Fig. 6 (9 & 15).

18. Line 262: 'traizole' should read 'triazole'.

-We have corrected the spelling.

19. The melting temperatures observed are not particularly elevated. I do not think it matters so much whether it is 8 or 30 K. I would perhaps moderate the wording.

-As per the suggestions we have revised the manuscript.

20. What is the rationale for the 'high' melting temperature induced by compound 21? The compound was not selected by in situ click with G4 if I understand correctly.

-The compound 21 was not obtained in TGS. Though the compound showed high melting temperature which may be largely due to the carboxamide side chain, its parent fragment alkyne 1b may not be able to bind to the DNA template to favour the cycloaddition reaction with azide 7.

Alkyne 1b has comparatively weak interaction with DNA due its lacking of electrostatic interactions. But once the compound 21 is formed (with Cu aided reaction), it showed high melting temperature due to its carboxamide side chain

21. Also, why the melting temperature induced by compound 10 is similar for both ds and G4 DNA while it was not obtained with the G4 in situ? What is the effect of this compound on MYC?

-In HPLC analysis we have obtained compound 10 in lower yield by c-MYC directed TGS and the control experiment with duplex DNA solely produced the compound 10. In FRET experiment the compound 10 showed similar binding stabilization for both G4-DNA and duplex DNA.

As per our knowledge, the HPLC chromatogram does not infer the binding affinity of the compound. It just indicates the product formation and percentage of yield. Though the compound showed comparable ΔT_m values for c-MYC G4 DNA, it does not mean that it would be generated in significant amount in TGS.

22. The scale bar in Fig 7a is missing. The quality of DAPI is not great. A data without DAPI would be welcome. As it is, we do not know what the effect of DAPI is on the nuclear staining of the compound alone. As mentioned above, this data shows a pan-nuclear staining that looks like the ds minor groove binder DAPI. This suggests that the compound interacts with many more DNA genes/G4/ds than the paper would have one to believe. Thus, the question is whether MYC is down-regulated as a result of direct targeting or something else, for example a global DNA damage induction that would also explain apoptosis.

We have given the scale bar in revised manuscript.

As we have explained earlier that the ligand itself has a higher fluorescence emission property, it shows the green colour covering the entire nucleus indicating its ability to permeate the nuclear membrane. Though compared to the DAPI staining, we have observed a punctate staining with Adduct 14.

It is evident from the several experiments that our molecule is selective to cMYC-G4 DNA over duplex DNA, however we cannot rule out the other possible pathways in this complex cellular system. Here we have just tried to explore one of the possible targets.

23. Annexin/PI should be backed up by PARP and Caspase 3 cleavage, biochemical markers of the cell death pathway. Also, the inhibitor ZVAD should rescue some level of viability in cells treated with the ligand.

-In this paper we have examined the initial biological relevance of Adduct 14. Definitely we are interested to invade the detailed mechanism in near future. To ensure the apoptosis mechanism of this ligand in HCT116 cells we have also performed the caspase3/caspase7 activation assay and we have got the desired and comparable result as demonstrated in AnnexinV/PI assay.

Responses to the Reviewers:

Reviewer #1 :

The authors report a recyclable platform for the target-guided synthesis of small molecule biological binders using magnetic nanoparticle-bound DNA. Using this method, the authors detail the discovery, identification, and in vitro validation of a small molecule inhibitor of the c-MYC G-quadruplex sequence, which can be used to induce apoptosis in cancer cells. While this platform has important implications on the field, the manuscript includes overly detailed, procedural descriptions of the experiments and, as a result, it is not of high enough quality for publication in Nature Communications in its current form. Major revisions would be required before it could be resubmitted for reconsideration in Nature Communications.

Specifically:

1. The abstract is very detailed and may benefit from a more concise description of the goals and experimental results.

-As per the requirement, we have concised and modified the abstract and the manuscript accordingly.

2. Again, the introduction would benefit from a broader description of the experimental goals (before delving into the experimental detail). Also, the last sentence of the introduction should more directly articulate the hypothesis of the work.

-We have modified the introduction.

3. Figure 1c,d: These particles do not appear spherical. The size distribution looks very broad. This may not affect the use of these particles as catalysts (as long as their catalytic ability is not being quantitatively measured), but it certainly calls into question whether each of the particles is in fact of the form Au@Fe₃O₄, as opposed to purely Au or Fe₃O₄. Energy dispersive x-ray spectroscopy (EDX) of the final particle products would be useful to obtain a complimentary validation of the structures claimed. This could be resolved by an initial magnetic separation/purification process. Also, how was the size of these particles determined ("The average size of Fe₃O₄ MNPs is 11 nm and after coating with Au, the average particle size is increased to 15 nm")? If this is done by EM analysis, this should be noted.

-We have prepared Fe₃O₄ NPs using literature procedure and characterized using TEM and UV spectroscopy. Au@Fe₃O₄ nanoparticles were prepared by citrate reduction of gold salt on Fe₃O₄ nanoparticles, the resulting mixture was extensively washed with deionized water to remove the free water soluble gold particles. The Au@Fe₃O₄ NPs were then isolated by magnetic decantation. The UV-Vis spectrum of Fe₃O₄ MNP showed no characteristic peak in the visible region, while the dispersion of Au@Fe₃O₄ NPs displayed a SPR absorption band ~ 535 nm. The EDX spectra of Au@Fe₃O₄ NP indicated the presence of both Fe and Au in the particles (Figure S1d, S.I.). The TEM images show that particles are nearly spherical.

The average diameter of the NPs was determined by TEM software Gatan Digital Micrograph and AFM software Image Processing.

It is important to mention that Fe₃O₄, Au@Fe₃O₄ and DNA coated Au@Fe₃O₄ exhibit paramagnetic properties and can be separated through magnetic decantation protocol using an external magnet.

4. Although the PAGE experiments may suffer from incomplete particle separation (i.e., any purely Au NPs would remain after magnetic purification), they could be utilized to confirm the surface functionalization. This should be addressed briefly in the main text and all of the experimental details should be transferred to the SI.

-Before the attachment of DNA, we have extensively washed the Au coated Fe NPs to remove the water soluble pure Au NPs as described in the SI. Thus there is least chance of surface functionalization of pure Au NPs as it is already removed.

As per the suggestion, the experimental details has been included in the revised S.I.

5. Figure 3: Schematics can be a concise way of communicating a scientific method or experiment; however, this schematic is overly cartoonistic. It would be helpful if there were a key that delineated the chemical structure symbolized by each shape. This key could also be present in Figure 2 for continuity.

-We have incorporated symbolized structures in the schematic representation since it is difficult to incorporate all the 14 azide and alkyne building blocks.

6. Page 7: The paragraph detailing the use of LiCl for lead compound separation should be shortened or transferred to the SI. Why were the results of unsuccessful HPLC experiments included ("lead compounds could not be identified as overlapping peaks were obtained in HPLC analysis")? It may be worthwhile to explore alternative mobile phases and columns in order to achieve peak separation. Also, why would one separation process lead to overlapping peaks in HPLC, while a different separation process would result in clearly identifiable peaks? This may suggest that your purification process is affecting the lead compounds.

-As per the suggestions, the comparative HPLC experiments of the separation processes are transferred to the S.I. and the figure 4 is also modified showing only successful results. The figure 3 is simplified showing the working protocol for carrying out the templated cycloaddition using the nanotemplates and their separation from the reaction mixture as well as the identification of the lead triazole compounds.

In the separation processes (Using LiCl/heat), the reaction supernatant contains all the azide alkyne building blocks along with the newly generated products. Therefore, overlapping peaks of all the compounds appeared in HPLC.

In the modified process, first DNA nanotemplates bound with newly generated triazole products were separated by an external magnet and washed to remove the unreacted azide and alkyne building blocks. The unreacted fragments remained in the supernatant. The MNPs were then re-suspended in buffer, heated to 65 °C for releasing the products bound with the DNA nano-template. The DNA-MNPs were separated instantly by magnetic decantation and the supernatant was collected and characterized by HPLC and ESI-MS analysis resulting identifiable peaks for the best binding triazole ligands.

We have used reverse phase column using water/acetonitrile as mobile phase because the templated cycloaddition is carried out in aqueous solution as well as for the isolation of water soluble lead compounds.

7. Page 12-14: These control experiments would be better included in the SI. They are very detailed and their results and implications can be summarized in a few sentences in the main text.

-The control experiments have been briefed in the manuscript and included in the S.I.

8. Conclusion: Although a control particle with double-stranded (as opposed to G quadruplex DNA) provides a valuable control, does the orientation of DNA on the surface of the NPs preferentially catalyze the formation of one lead compound over another? How does the distribution of lead compounds catalyzed by particle-bound DNA compare to free DNA? Is this orientation potentially beneficial (e.g., do the lead compounds catalyzed by the oriented DNA bind more strongly than those produced by free DNA)? While this platform provides an interesting method for reusing catalysts, the manuscript may benefit from a more chemical description of the benefits of the surface-bound DNA.

We have examined the templated cycloaddition using the same azide-alkyne library in the presence of free G-quadruplex DNA. As we have mentioned in the article, similar lead compounds **Tz 1** and **Tz 3** were obtained. However, as expected, we have observed overlapping spectra for the products along with unreacted azide-alkyne fragments; so we could not monitor the exact distribution of the lead compounds.

As per our knowledge, the orientation of the NP bound DNA and the free DNA did not have any effect on the formation of lead compound (evidenced by CD spectroscopy, Fig 1c). Here, the DNA was immobilized on the nanoparticle support for the purpose of recycling the same DNA molecule. The DNA did not change its folded conformation upon attachment on the NP surface as indicated by CD spectroscopy. Therefore, both of them would generate similar product distribution.

Also it DNA immobilization on NP surface would provide a large surface area with numbers of localized nano-template centres that would speed up the reaction rate and produce a higher yield.

9. The authors show strong analytical support for most of the claims in the main text of the manuscript. In general, this manuscript would be greatly strengthened by highlighting the key aspects and conclusions of each experiment. In its current form, these higher level ideas are lost in the incredibly detailed text.

-We have modified the manuscript by highlighting our main focus in this paper.

Reviewer #2 (Remarks to the Author):

The MS by Panda et al. combines several elements for the selection of c-MYC G-quadruplex (MYC-Q) binders. Gold covered magnetic nanoparticles are synthesized and the MYC-Q are immobilized on the nanoparticles via thiol chemistry. The main purpose of the study is to use the MYC-Q on nanoparticles as the target in Target Guided Synthesis. It is a method where the binding of building blocks from a library to the target templates the formation of covalent bonds between building blocks, in this case an alkyne and an azide. In this manner highly selective binders may be formed from libraries of building blocks. A set of building blocks containing 2 alkynes and 7 azides are used in the present study and 3 binders of which 2 are specific are identified. It is verified that the binders have high affinity for MYC-Q and that they down-regulate c-MYC expression and induce apoptosis in cancer cells.

While many aspects of the work are interesting I also have some reservations:

It is claimed in the title and through the MS that the MYC-Q is a catalyst for the azide-alkyne reaction of selected substrates. I see no evidence for a turnover >1 and I would say that MYC-Q serves as a template for the reaction rather than a catalyst.

-We have mentioned DNA nanoparticles as template rather than catalyst in our revised manuscript.

The authors also promote in the title and elsewhere that by immobilizing MYC-Q on the nanoparticles, the MYC-Q is recyclable. However, only one round of recycling is demonstrated and the yields of the products are changed after the first round. Several rounds (>10) should be demonstrated if the recyclability should be of any value.

-We have examined the recyclability of the system for 3 reaction cycles and the product distribution are more-or-less similar. However, it is noteworthy that the main lead compound is formed as a major product yield in every recycling process.

Additionally we have performed the DNA templated synthesis in a time dependent manner (2, 4 and 6 days), which shows that the DNA template preferentially generates strong binders in a shorter time scale.

My most critical reservation is that the library is too small with a total size of only 14 possible combinations. The authors do find good binders, but the fragments are very similar to fragments of compounds that have previously been shown to bind to MYC-Q by the same authors. The identification of the binders, still happens by HPLC and MS and hence it is difficult to see how this should be expanded even to medium-sized libraries of >1000 combinations. The parallel testing of all combinations in individual vials is much easier in spite of the cost of DNA, especially since the product is fluorescent and can immediately be identified. Therefore I have a hard time seeing the potential value of the method.

-In the revised manuscript, we have increased our azide and alkyne building blocks that would generate 66 possible triazole products including 1,4 and 1,5 regioisomers. Our results show that G-quadruplex DNA nanotemplate could selectively form a major triazole lead compound.

First, the synthesis of all 66 potential compounds and testing their individual binding affinity using biophysical analysis are time-consuming and laborious. Secondly, the individual screening procedure requires different biophysical methods that consume a large amount of DNA in each experiment.

The advantage of utilizing the recyclable DNA templates in our procedure over the normal screening is to combine both the synthesis and screening procedure in one step that allows efficient isolation of both lead compounds and the DNA templates, consequently reducing the consumable expenses and time. This method is able to generate a specific ligand for a particular target in a better way than the conventional methods.

In conclusion I do not find that the MS is sufficiently important to be published in Nature Communications.

This work also delineates the first example of a NP-immobilized DNA for templating TGS for the identification of the best binder from azide and alkyne fragments leading to regioselective formation of the triazole ligands. The use of NP as solid support to template the Huisgen reaction is novel. The generated triazole ligand has the potential to inhibit c-MYC expression in cellular system by directly targeting the c-MYC quadruplex. So far, only one compound (a benzofuran derivative, *ACS Chem. Biol.* 2016, 11, 139–148) is reported in the literature showing MYC inhibition in a G4- dependent fashion. These results are included in our revised manuscript.

Reviewer #3 (Remarks to the Author):

Dash and co-workers report on the development and use of gold nano-particles coated with nucleic acids folded into either double stranded or G-quadruplex DNA structures. These DNA-NPs can be used as catalysts to template the formation of G-quadruplex specific ligands. In particular, the authors are using these catalysts to mediate Huisgen cycloadditions between a library of alkynes and azides with the view to select for the best binder. By doing so, the authors identified a small number of ligands that can stabilize the G-quadruplex motif that can be found in the promoter of MYC oncogene. Then, the authors show that this compound, which is fluorescent, enters cell nuclei and down-regulate MYC at the mRNA and protein levels. The authors show that a short-term treatment induces apoptotic cell death and conclude that this phenotype occurs as a direct consequence of MYC promoter targeting.

The paper is clear and well-written. The authors have used a large diversity of techniques in this study, which is not very common and quite impressive. This include material sciences, supramolecular chemistry, synthetic organic chemistry, confocal microscopy, AFM, TEM, circular dichroism, FRET-melting assays, PCR, western blotting, apoptosis assay. Most of the important controls are also reported. While the work is interesting, I do have some concerns that are listed thereafter.

Major concerns:

1. While it has previously been reported that small molecules can target the G4 in MYC and induce a down-regulation of this gene, there is some scepticism in the field. As yet, no solid proof that the down-regulation of MYC induced by small molecules has anything to do with this structure. There is no doubt that the structure described can form a G4 and can be stabilized by small molecules, as described in this study. However, the concentrations of drugs used to detect significant changes in cells are too high and the effect likely arises as a result of off-target effects. In particular, the authors show by microscopy that the small molecule targets the entire nuclear area. While there are only two copies of the MYC G4 structure, how can the author possibly claim that cell death and MYC down-regulation results from the direct targeting of this structure? The authors are not to be blamed. The entire field publishes such articles and in my opinion, if it is clear that G4 structure in cells have been demonstrated by several labs, in particular by the Balasubramanian laboratory, the existence of isolated, functional/druggable motifs in the promoter region remains controversial and unsubstantiated.

-We have given subsequent experimental proofs (Western blot, mRNA) along with the biophysical data which indicates that **Tz 1** stabilizes *c-MYC* quadruplex DNA and downregulates *c-MYC* expression both in transcriptional and translational level.

For further validation, we have performed "CA46 exon-specific" assay, which is presently the only cellular assay that can discriminate between repression of *c-MYC* expression through direct binding to the G-quadruplex in the *c-MYC* promoter versus a secondary ligand effect. In this assay, we have observed that **Tz 1** significantly induced exon-specific downregulation of *MYC* expression in a dose-dependent manner. These results are included in the revised manuscript.

In recently published papers, the G-quadruplex binding ligands show the IC₅₀ value ranging from 0.5-10 μM (*J. Med. Chem.* 2011, 54, 5671–5679, *ACS Chem. Biol.*, **2016**, 11,139–148). In our paper, the IC₅₀ value of **Tz 1** in HCT116 cells (2.1 μM) is comparable with those peer reviewed publications. And thus, we have performed other experimental assays below the apoptotic dose (< 2 μM).

The compound **Tz 1** itself is highly fluorescent molecule. The fluorescence microscopic images of HCT116 cells treated with **Tz 1** indicated that the molecule localizes in the nucleus and transmit its own fluorescence upon excitation and thus the entire nucleus turns green. In our opinion it is very difficult to distinguish the fluorescence difference between the compound bound to G-quadruplex DNA and the free compound by fluorescence microscopy. This experiment only indicates the permeability of the ligand through the nuclear membrane which is a basic necessity of a DNA binding druggable compound.

2. The use of NP as solid support to catalyze the Huisgen reaction is novel. However, as stated by the authors, the use of nucleic acids and G4 structures in particular to catalyze this reaction and identify potent binders is not novel. See ref 19 and 20 of this manuscript. It seems that the methodology described here is rather similar to that previously reported by Antonio et al in ref 19. It is not clear what is really novel in this article?

- The majority of TGS approaches, reported to date, use various enzymes as the target to assemble their potent inhibitors. Only two examples are reported so far using nucleic acids as the targets. However, these solution phase TGS methods have some limitations like poor isolation of the lead compounds from the reaction mixture

comprising the target and fragment library and lack of reusability of the target for multiple rounds of templated synthesis.

The solution phase cycloaddition with free G-quadruplex DNA has been reported by Antonio et al and Hu et al using different G-quadruplex targets (h-telo, c-kit, h-ras DNA and a RNA quadruplex). Our study used *c-MYC* quadruplex as the template for the first time to discover a quadruplex dependent potent MYC inhibitor.

Our study further provides several advantages over the published methods by using this DNA-NP system:

First, (i) we have prepared and characterized magnetic nano supported G-quadruplex and duplex templates and used them as targets to template metal free cycloaddition for the first time.

Secondly, (ii) we expected that DNA templates would provide a larger surface area with more numbers of reaction centres to facilitate the reaction between small molecule fragments with complementary reacting groups. Our time dependent TGS experiments demonstrate that DNA templates can selectively assemble the selective and major lead compounds in a shorter time scale.

Third, (iii) the lead compounds can be efficiently isolated by simple heating followed by magnetic decantation. Moreover, the system enables the recycling of the DNA templates and consequently reduces time and cost.

This work therefore delineates the first example of a nanoparticle supported TGS for the identification of potent regioselective triazole ligands over the existing solution TGS methods using nucleic acid as the targets.

We believe that this recyclable nano-template approach can also be applicable with other biomolecules other than G-quadruplex DNA to easily identify the target specific binders.

Minor concerns:

1. What is the link between MYC inhibition and induction of apoptosis? Unless cancer cells are addicted to this oncogene, it is not obvious why taking out a transcription factor would initiate cell death. What is the effect of a knock-down of this gene?

As per the literature, the *c-MYC* gene is over expressed in HCT116 cells (*J. Med. Chem.* **2012**, *55*, 6076–6086) and this is one of the hallmark of cancer (*Oncogene* **2008**, *27*, 6462–6472). There are several evidences where inhibition of *c-MYC* expression induces apoptosis (*Mol. Cell. Biol.* **1999**, *19*, 1-11, *Oncogene* **1999**, *18*, 2967- 2987) in cancer cells.

Over expression of a major transcription factor may lead to the abnormal cell growth as well as inhibition of the expression may rescue the normal growth to some extent. Knockdown of *c-MYC* may lead to the inhibition of the cell growth and gradually may lead to the cell death (*Breast Cancer Res.* **2005**, *7*, R220–R228).

Herein we have demonstrated that the lead compound **Tz 1** is able to bind the *c-MYC* G-quadruplex DNA, reduces the expression of *c-MYC* oncogene and subsequently induces apoptosis in HCT116 cells.

2. One advantage of the method appears to be the fact that the catalyst can be recycled. Is this really important? Oligonucleotides are not that expensive. This is a distraction in the paper. The authors should keep the focus on catalysis, turn-over, molecular recognition...

-We have removed the recycling of DNA templated from the title. We have highlighted that nanoparticle supported DNA sequences can be used to assemble their potent ligands from a library of reactive azide-alkyne fragments based on molecular recognition. We have used the term template for the DNA targets instead of catalyst.

The use of DNA linked nano-particles as the targets not only facilitate efficient isolation of the lead compounds but also promote the recovery and reuse of the DNA-nanotemplates. In this context, we have mentioned the recycling DNA NPs that may reduce the expenses of commercial DNAs, required in high amounts to perform conventional screening assays.

3. Is the probe really a turn on probe or is it just fluorescent? If it is a turn on probe, is it upon binding to G4 or ds DNA? What are the consequences on the interpretation of the confocal microscopy data?

Ligand **Tz 1** is itself a fluorescent molecule. In our experiment, we found that **Tz 1** shows a 5 fold increase in fluorescence intensity with a 33 nm blue-shift in emission maxima in the presence of *c-MYC* G-quadruplex DNA (Figure 6b). In comparison, we did not find any significant increase in fluorescence with dsDNA (Figure 6c). In this respect, **Tz 1** can be considered as a "turn on" probe in vitro.

The cellular system is much more complex and diverse compared to the in vitro system. Binding cannot be firmly concluded only from the microscopic image. Our goal is to show that the ligand permeabilizes the nuclear membrane and localizes into the nucleus after entering into the cell, which is an obvious criteria for a DNA binding drug.

4. In relation to my previous point, the authors claim that the effect is linked to the targeting of a single G4 motif while the targeting appears to be pan-nuclear. Another, and in my opinion more relevant, angle would be that this molecule is a pan-G4 binder, which would explain why the staining is pan-nuclear. There is no way this compound can be selective to the G4 in MYC and not target the other 750.000 identified by sequencing (see Nat Biotech, Balasubramanian et al cited in this paper). In particular, this compound could well target all these motifs and induce DNA damage in a replication/transcription-dependent manner. This would explain the apoptosis reported by the authors. This has previously been described by few groups including the Jackson lab (Nat Chem Biol 2012). The authors should look at some DNA damage marks after a short period treatment and see whether the DDR is activated. This should be done at a time point that precedes apoptosis, of course.

-In this paper we are particularly interested to evaluate the binding selectivity of the target guided synthesized compound towards quadruplex DNA over duplex DNA. Here *c*-MYC just represents one type of quadruplex DNA.

5. The authors are highlighting the value of their method, which is very fair. However, TGS with DNA has been reported. The authors here claim that it takes 6 days to select for small molecules using G4-NP in absence of copper I assume. The ref 19 describes similar reaction times. Thus, the NP strategy does not appear to have a much higher turn-over. What happens with lower amounts of NP and shorter time period? To the very least, the authors should compare the methods experimentally to make such a claim. This could, in fine, demonstrate the advantage of NP vs solution TGS with nucleic acids.

-As per the suggestion, we further carried out the TGS reaction with $G_4 \bullet Au @ Fe_3O_4$ template at two different time points. We have obtained the major lead **Tz 1** and the duplex DNA binder **Tz 2** after 2 days of reaction. After 4 days, the formation of **Tz 1** was increased compared to **Tz 2**, while the other lead **Tz 3** was not detected in the mixture. It is important to mention that the formation of **Tz 1** was greatly accelerated by G-quadruplex DNA template with increasing time. However, the formation of **Tz 2** was not significantly increased by the G-quadruplex template with time (Figure S7, S.I.).

The time dependent TGS with $G_4 \bullet Au @ Fe_3O_4$ template indicated that the DNA template preferentially generates strong binders in a shorter time scale. This further points out the advantage of NP vs solution TGS with nucleic acids which permit rapid identification of lead compounds as well as allows the reuse of the DNA template.

6. Is there any selection with a mutated MYC sequence that cannot form a G4?

-In the manuscript, we have used a *c*-MYC quadruplex nano template and a duplex nano template, for identifying selective ligands for quadruplexes over the duplex DNA.

We have also explored the binding ability of this ligand in a single stranded non-quadruplex DNA (apart from dsDNA) and not observed any significant changes in fluorescence titration (results were not included in the manuscript).

7. What is the value of using TEM and AFM data? Is it really important to have homogeneous mono-disperse material for an effective catalysis (which is the main point of this paper if I understand correctly)?

-The homogenous mono-disperse material is not required for target guided synthesis. We have shown the data for the characterization of the NPs.

8. The CD spectra in SI are not normalized on the y-axis. CD/mdeg is not a unit and does not take into account the experimental settings including concentration of the oligos.

-We have followed the experimental methods as per the mentioned publications. (*Chem. Commun.*, **2012**, *48*, 7607-7609, *Nucleic Acids Res.* **2012**, *40*, 10334-44)

9. Figure S2. The gel contains no control oligo ladders. The control used shows several bands? Also, the quality of the gel presented is not great and could be improved.

-The control lane showed several bands in this 20% Native PAGE that may correspond to the different folded conformations of *c*-MYC G-quadruplex DNA.

10. Did the authors attempt to add a copper catalyst in addition to the G4-NP? While the NP could promote the correct selection, one could expect that copper reduces the time of the reaction (perhaps at the expense of the selection).

- Theoretically addition of copper speeds up the reaction but generates all the possible 1,4-triazoles and the true ligand may not be distinguished.

We intend to know the type of the regio-isomer formed in the presence of metal free DNA templated cycloaddition. In this study, DNA nano-template produced a 1,4 triazole ligand and we did not detect any 1,5-product. Therefore, we have synthesized the lead ligands in large scale by copper catalysed cycloaddition reaction for further evaluation purposes.

11. In terms of molecular recognition, it is indeed conceivable that the hetero-aromatic carbazole interacts by means of pi-pi interactions but the surface covered by this fragment is rather small and could equally interact with a single DNA base. Thus, the ds-DNA could in principle also select for good binders. Is this happening? If not, what is the rationale behind this. If it does, are the selected compounds interacting with genomic DNA in live cells and what is the effect?

-The individual alkyne and azide fragments can weakly interact with both G-quadruplex DNA and dsDNA. In TGS, only those fragments come in proper orientation on binding with G-quadruplex or ds DNA that lead to the formation of potent target binding ligands. In our case we have obtained 3 leads for G-quadruplex DNA and only one of them was obtained for dsDNA.

The major lead compound **Tz 1** contains two protonable tertiary amine side chains (derived from alkyne **1a** and azide **11**, both contain positively charged amine side chains) can strongly interact with quadruplex compared to the second lead compound **Tz 3** (derived from alkyne **1a** and azide **7** that lacks positively charged tertiary amine side chains) that contains one amine side chain. In addition, the minor lead **Tz 2** (derived from alkyne **1a** and azide **3**) that lacks the aromatic carboxamide group was formed as a major product in the presence of duplex DNA template, which further indicates that compounds with extended aromatic system can selectively target G-quadruplex over duplex DNA. The triazole compound **Tz 1** that showed high affinity for the *c-MYC* G-quadruplex DNA contains both extended π -system for efficient overlap with G-quadruplex and two positively charged amine side chains for strong interaction with the sugar phosphate backbone of G-quadruplex.

12. Same comment for electrostatic interactions. The phosphate backbone can be found also in ds-DNA. Why should these molecules selectively interact with the G4?

-We have obtained **Tz 2** by dsDNA templated TGS. As we have mentioned above, here the side chains play key role in determining the selectivity but only when they are conjugated in the triazole product. The amine group of azides **3** and **11** is responsible for the electrostatic interaction with the phosphate backbone. So azide **3** derived compound **Tz 2** non-selectively binds both G4-DNA and dsDNA. But the conjugated aromatic ring of azide **7** and **11** further involved in stacking interaction with the external G-quartet of G-quadruplex and also prevent the non-selective interaction with dsDNA (compound **Tz 1** and **Tz 3**).

13. While it might be obvious, the authors should explain their choices of fragments employed. Since the library is rather small (it does not need to be large), it may be wise to comment on types of supramolecular interactions these functions can engage rather than the functional groups themselves.

-We have increased the library size and we have also discussed the reason for selecting the fragments in the revised manuscript.

14. Heating at 65°C (even with the presence of LiCl) may not be stringent enough to release the binders from the NP, especially if the ligands identified are tight binders. This can introduce biases towards the analysis, or what appears to be the likely outcome of the selection process. How can one be sure that the compounds have been quantitatively recovered from the NPs?

-At high temperature, the thiolated DNA molecules can be detached from the nano-surface (*Colloids Surf B Biointerfaces*. **2006**, *51*, 130-9). And therefore, if we apply temperature higher than 65 °C, the DNA templates can not be reused. We repetitively performed the separation until complete isolation of the ligand as confirmed by HPLC.

In a control experiment, we have denatured the DNA nanotemplate at higher temp. (95 °C) to obtain similar distribution of products.

15. The authors mention that Li prevent recycling of the NP. Can the G4 not be re-folded in the presence of potassium? If not, what are the reasons?

- Our protocol was as follows: By treating the reaction mixture with 8M LiCl followed by heating to 65 °C to separate the triazole products from the DNA, we have observed an inseparable complex mixture of compounds with overlapping peaks in the HPLC chromatogram of the supernatant. The MS analysis however showed that the supernatant contained a mixture of newly generated triazole products and unreacted azide and alkyne building blocks. The DNA nanoparticles were separated by applying simple magnetic field.

The separated G4•Au@Fe₃O₄ MNPs could not be reused for another round of azide-alkyne cycloaddition as Li ions destabilizes the quadruplex confirmation. It might be possible to again refold the G-quadruplex template in potassium, but the method is not suitable for the identification of the triazole products using HPLC.

Since we have modified the separation protocol for efficiently isolating the lead products and DNA templates, we did not subsequently use LiCl for the separation.

16. One could use, as previously described, a biotinylated oligo, and recover the ligands similarly. It would be more simple. What do gold/thiol provide over biotin/streptavidin?

-The use of biotinylated oligo (expensive) is also simple, but our system is quite new and advantageous because of the following reason.

The system, as we have mentioned in the manuscript, provides a number of recyclable reaction centres on a single surface that would speed up the TGS which is not possible with the biotinylation.

17. What happens when the fragments leading to the lead compounds are removed from the selection process? Did the authors detect other, less efficient, binders? If so, are these still biologically active?

-The combination of alkyne 1a and azide 3, 7 and 11 gives the lead compounds (Fig. 2 and Fig. 4) amongst which the triazole product of alkyne 1a and azide 11 has been generated in higher percentage. Since we did not obtain any triazoles derived from alkyne 1b and 1c, thus we may confer that in the absence of alkyne 1a, there will be no significant product.

18. Line 262: 'traizole' should read 'triazole'.

-We have corrected the spelling.

19. The melting temperatures observed are not particularly elevated. I do not think it matters so much whether it is 8 or 30 K. I would perhaps moderate the wording.

-As per the suggestions we have revised the manuscript.

20. What is the rationale for the 'high' melting temperature induced by compound 21? The compound was not selected by in situ click with G4 if I understand correctly.

-As per suggestion, we have increased the library size and therefore we have shown the FRET melting analysis of only the three lead compounds.

21. Also, why the melting temperature induced by compound 10 is similar for both ds and G4 DNA while it was not obtained with the G4 in situ? What is the effect of this compound on MYC?

-In HPLC analysis we have obtained compound 10 (now **Tz 2**) in lower percentage by c-MYC directed TGS and the control experiment with duplex DNA solely produced the **Tz 2**. FRET and fluorescence binding titrations reveal similar binding affinity of **Tz 2** for G-quadruplex and duplex DNAs indicating **Tz 2** as a non-selective probe. The G4 DNA templates the formation of **Tz 1**, **Tz 2** and **Tz 3** in 62:7:31 percentage ratio, the compounds **Tz 1** and **Tz 3** that formed as major products exhibited higher binding affinities for G4 DNA over **Tz 2**.

Since **Tz 2** non-selectively binds both duplex and quadruplex, we have not used this ligand in biological system.

22. The scale bar in Fig 7a is missing. The quality of DAPI is not great. A data without DAPI would be welcome. As it is, we do not know what the effect of DAPI is on the nuclear staining of the compound alone. As mentioned above, this data shows a pan-nuclear staining that looks like the ds minor groove binder DAPI. This suggests that the compound interacts with many more DNA genes/G4/ds than the paper would have one to believe. Thus, the question is whether MYC is down-regulated as a result of direct targeting or something else, for example a global DNA damage induction that would also explain apoptosis.

We have given the scale bar in revised manuscript.

As we have explained earlier that the ligand itself has a higher fluorescence emission property, it shows the green colour covering the entire nucleus indicating its ability to permeate the nuclear membrane. But we have observed a punctate staining with **Tz 1** compared to the DAPI staining that stains the whole nucleus.

It is evident from the several experiments that **Tz 1** is selective to *c-MYC*-G4 DNA over duplex DNA and CA46 exon specific assay further concludes that the observed *c-MYC* downregulation is a direct effect of **Tz 1** mediated G-quadruplex stabilization.

23. Annexin/PI should be backed up by PARP and Caspase 3 cleavage, biochemical markers of the cell death pathway. Also, the inhibitor ZVAD should rescue some level of viability in cells treated with the ligand.

-In this paper, we have examined the initial biological relevance of **Tz 1**. Definitely we would like to investigate the detailed mechanism in near future. To ensure the apoptosis mechanism of this ligand in HCT116 cells, we have performed the caspase3/caspase7 activation assay and we have obtained the desired and comparable results (unpublished data) as observed in AnnexinV/PI assay.

Reviewer #1 (Remarks to the Author):

The authors report a recyclable platform for the target-guided synthesis of small molecule biological binders using magnetic nanoparticle-bound DNA. Using this method, the authors detail the discovery, identification, and in vitro validation of a small molecule inhibitor of the c-MYC G-quadruplex sequence, which can be used to induce apoptosis in cancer cells. While the authors have made significant improvements to the clarity of the manuscript and added additional insights, there are still a few concerns that should be addressed prior to publication. Specifically:

1. “the DNA sequences functionalized with NPs” (line 43-44): Do the authors mean the NPs functionalized with the target DNA strand?

2. “the DNA sequences functionalized with NPs would greatly enhance the rate of reaction between small molecule fragments by providing a larger surface area with more numbers of localized DNA templates” (line 43-45): The authors do not provide evidence that this is the case either in terms of the yield or the amount of target produced. (“For a comparison, we have performed the TGS reaction with conventional c-MYC G135

quadruplex DNA sequence (5'-G4AG3TG4AG3TG4-3'), that yielded essentially similar lead compounds” (lines 134-135)).

3. In response to the original comment #8, the authors included an argument that “the DNA immobilization on NP surface would provide a large surface area with numbers of localized nano-template centers that would speed up the reaction rate and produce a higher yield.” While this may be true, the authors do not provide evidence that the nanoparticle template increases the reaction rate in the text.

Reviewer #2 (Remarks to the Author):

The authors have improved the MS in the revision, however I still have some serious concerns:

1) It is surprising how the size of the library increases from 14 to 66 by the addition of one alkyne and four azide building block. The considerations about the formation of the 1,5-triazole are speculative. It is well-know that the 1,5-isomer is formed at elevated temperatures, but at room temperature it is only obtained with very specialized catalysts. Therefore, unless the authors can

identify the 1,5-isomer from the NP-templated reaction I would recommend that they leave out that part and sets the library size to 33 possible isomers. It is still a very small library.

2) I do not find any experimental evidence for the recycling and conservation of templating activity after recycling of the nanoparticles.

3) A new concern is related to that fact that only alkyne 1a which contains an aliphatic amine is observed in the isolated products. Are alkynes 1b and 1c at all soluble in the reaction buffer? In which formulation are they added to the reaction? If they simply do not dissolve/precipitate it is obvious that they are not observed in the product.

Furthermore, the products are identified from the supernatant i.e. from an aqueous solution, which means that they have to be water soluble in that buffer, and strikingly all products contain one or more amines that are protonated at neutral pH. Many of the product combinations are certainly not soluble in aqueous buffer and would most likely not be observed in the supernatant. Could it be that the selectivity is mainly caused by the solubility of the reactants and products?

Reviewer #3 (Remarks to the Author):

The authors have only partly addressed the comments raised by the reviewers. For example, CD spectra have not been normalized as indicated, no rationale linking Myc suppression and apoptosis has been provided, the fact that the selected ligands appear to be pan-genome binders does not seem to be a concern for the authors, who claim that Myc down-regulation arise as a consequence of direct targeting of a G4 motif in the promoter of this gene but no evidence (e.g. pull-down/sequencing) as been provided. There are other G4 motifs in the gene body of Myc that could equally be targeted by these compounds and lead to the same result. This possible scenario has not been discussed. Also, the selection concept is not novel and this study represents only an improvement of other methodologies previously reported. Thus, while the work is of interest, it may not reach the standard of Nature Communications.

Response to Reviewers letter

Reviewer #1 (Remarks to the Author):

The authors report a recyclable platform for the target-guided synthesis of small molecule biological binders using magnetic nanoparticle-bound DNA. Using this method, the authors detail the discovery, identification, and in vitro validation of a small molecule inhibitor of the c-MYC G-quadruplex sequence, which can be used to induce apoptosis in cancer cells. While the authors have made significant improvements to the clarity of the manuscript and added additional insights, there are still a few concerns that should be addressed prior to publication. Specifically:

1. “the DNA sequences functionalized with NPs” (line 43-44): Do the authors mean the NPs functionalized with the target DNA strand?

- We have revised in the manuscript as the DNA sequences immobilized on the nanoparticle surface.

2. “the DNA sequences functionalized with NPs would greatly enhance the rate of reaction between small molecule fragments by providing a larger surface area with more numbers of localized DNA templates” (line 43-45): The authors do not provide evidence that this is the case either in terms of the yield or the amount of target produced.

(“For a comparison, we have performed the TGS reaction with conventional c-MYC G135 quadruplex DNA sequence (5'-G4AG3TG4AG3TG4-3'), that yielded essentially similar lead compounds” (lines 134-135)).

-When the TGS was performed with conventional c-MYC G-quadruplex DNA sequence, an overlapping peak was obtained in HPLC chromatogram (Figure 1a). However similar products were identified in ESI-MS. To avoid the unnecessary complexity we tried to determine the reaction rate using a single pair of alkyne and azide (alkyne **1a** and azide **11**).

We have performed the TGS with c-MYC G₄•Au@Fe₃O₄ nanoparticle template in a time dependent manner with the above-mentioned combination of alkyne-azide and calculated the relative yield from HPLC chromatogram (Figure 2). After 4 days, we obtained 44% yield of **Tz 1**.

Figure 1: TGS with free G4-DNA.

Figure 2. The rate curve of TGS with *c*-MYC G₄•Au@Fe₃O₄.

The TGS was next performed with free G4-DNA with the same alkyne azide combination for a comparison. However, we have obtained a single peak for *c*-MYC G4-DNA along with bound **Tz 1** (Figure 1b). Further, we have digested the *c*-MYC DNA with DNase I to discriminate **Tz 1** from *c*-MYC DNA. We get a distinct peak for **Tz 1** (confirmed by ESI-MS of the collected fraction) smaller in size compared to our method (Figure 1c). We therefore feel that it did not represent the total **Tz 1** formed by free G4-DNA. This may be due to the incomplete DNase I digestion because of the strongly bound **Tz 1**:G4-DNA complex. Therefore it was difficult to determine the relative yield and rate of this TGS reaction.

Further, the TGS using free DNA and the nanotemplate may not be compared as their loadings are different. We are using very minute nanoparticles that are loaded with low concentration of DNA (nearly 6-7 μ M conc.) for the nano-TGS compared to free DNA (50 μ M conc. is used).

3. In response to the original comment #8, the authors included an argument that “the DNA immobilization on NP surface would provide a large surface area with numbers of localized nano-template centers that would speed up the reaction rate and produce a higher yield.” While this may be true, the authors do not provide evidence that the nanoparticle template increases the reaction rate in the text.

As discussed in the previous response, the relative rate of the reaction has been calculated (Figure 2). Since the reaction rate of the TGS reaction with free G4-DNA could not be calculated, the comparative study was not performed.

Reviewer #2 (Remarks to the Author):

The authors have improved the MS in the revision, however I still have some serious concerns:

1) It is surprising how the size of the library increases from 14 to 66 by the addition of one alkyne and four azide building block. The considerations about the formation of the 1,5-triazole are speculative. It is well-known that the 1,5-isomer is formed at elevated temperatures, but at room temperature it is only obtained with very specialized catalysts. Therefore, unless the authors can identify the 1,5-isomer from the NP-templated reaction I would recommend that they leave out that part and set the library size to 33 possible isomers. It is still a very small library.

-We have included 3 alkynes and 11 azides in the library which can give rise to 66 product combinations including 1,4 (33 triazole products) and 1,5-regioisomers (33 triazole products). In the previous literature of Sharpless (*Angew. Chem. Int. Ed.* **2002**, *41*, 1053) and Kolb's group (*J. Am. Chem. Soc.* **2004**, *126*, 12809-12818), the acetylcholinesterase enzyme was used as a test system in target guided synthesis that has generated 1,5-disubstituted triazole product instead of 1,4-regioisomer. So, we have described the possible generation of 66 triazole products that are the combinations of alkyne 1a-c and azide 2-12. In addition, there are several recent literatures that has considered the possible generation of both 1,4 and 1,5 triazole products from the azide-alkyne library in target guided synthesis [*Angew. Chem. Int. Ed.* **2012**, *51*, 11073; *Nat. Comm.* **2017**, *8*, (doi: 10.1038/s41467-016-0009-6)].

2) I do not find any experimental evidence for the recycling and conservation of templating activity after recycling of the nanoparticles.

-We have now included the evidence for the recycling and conservation of templating activity of G₄-Au@Fe₃O₄ nanoparticles in supporting information (Fig S12). Even after 3rd cycle of using the same template, **Tz 1** was obtained which indicates the efficient recycling ability of c-MYC G₄•Au@Fe₃O₄ nanoparticle template. This part has been discussed in the main manuscript.

3) A new concern is related to that fact that only alkyne 1a which contains an aliphatic amine is observed in the isolated products. Are alkynes 1b and 1c at all soluble in the reaction buffer? In which formulation are they added to the reaction? If they simply do not dissolve/precipitate it is obvious that they are not observed in the product.

We have prepared a DMSO stock solution of every building blocks and then their working concentration were prepared by serial dilution in Tris.KCl buffer with 5% DMSO. In this formulation, neither the building blocks get precipitated in aqueous solution nor do they get any solubility advantage in this TGS.

4) Furthermore, the products are identified from the supernatant i.e. from an aqueous solution, which means that they have to be water soluble in that buffer, and strikingly all products contain one or more amines that are protonated at neutral pH. Many of the product combinations are certainly not soluble in aqueous buffer and would most likely not be observed in the supernatant. Could it be that the selectivity is mainly caused by the solubility of the reactants and products?

-The products obtained in this TGS contain amine side chains. The probable reason is that the amines show binding affinity towards DNA through electrostatic interactions. But it should be noted that one of the product (derived from azide 2) was not formed in TGS though it contains an amine group. Thus, we assume that the results obtained were not affected by any solubility problem in aqueous solution. Further the combination of alkyne **1b** with the azides **3**, **7** and **11** (which were selected by the DNA templates) would have produced water soluble triazole derivatives, but we did not observe the formation of these products using the DNA templates.

We have also analyzed the whole reaction mixture by Mass Spectroscopy and did not observe the formation of other triazole products, which suggests that DNA templates select the best binding ligand.

Furthermore, we have first prepared a solution of azide and alkyne building blocks and then added the nanotemplates and we did not observe the formation of any precipitates.

As per the comment, we have taken the nanoparticle template in Tris.KCl+5% DMSO solution after 6 days reaction and heated to collect the generated triazole product and obtained the same result.

In our previous draft of this manuscript, we have reported the synthesis of 14 triazole products (derived from alkyne **1a** and **1b** and azides **2**, **3**, **7**, **9**, **10**, **11**, **12**) and studied their binding stabilization for the *c-MYC* G-quadruplex and duplex DNA. Our results showed that the compounds **Tz 1** and **Tz 3** strongly interacted with the *c-MYC* quadruplex compared to other compounds. **Tz 2** was found to interact with duplex DNA. These results further validate that DNA templates select the strong binders by selecting appropriate azide and alkyne partners.

Reviewer #3 (Remarks to the Author):

The authors have only partly addressed the comments raised by the reviewers. For example, CD spectra have not been normalized as indicated, no rationale linking Myc suppression and apoptosis has been provided, the fact that the selected ligands appear to be pan-genome binders does not seem to be a concern for the authors, who claim that Myc down-regulation arise as a consequence of direct targeting of a G4 motif in the promoter of this gene but no evidence (e.g. pull-down/sequencing) as been provided. There are other G4 motifs in the gene body of Myc that could equally be targeted by these compounds and lead to the same result. This possible scenario has not been discussed.

As per the suggestion, we have now included the normalized CD spectra plot in Figure 1c and S17.

We thank for the advice and appreciate the concern raised. Several evidences have been reported in literature regarding the ligand-quadruplex interaction mediated *c-MYC* suppression and apoptosis (*ACS Chem Biol*, 2016, **11**, 139-148, *J. Med. Chem.*, 2017, **60**, 1292–1308, *J. Med. Chem.* 2011, **54**, 5671–5679). As per the standard protocols, we have demonstrated the carbazole derivative mediated G-quadruplex stabilization and subsequent *MYC* downregulation followed by apoptosis in cancer cells.

Though we have not performed any pull down or sequencing experiment but CA46 exon specific assay is currently the only method to illustrate the G-quadruplex dependent *c-MYC* gene regulation (*J. Biol. Chem.*, 2011, **286**, 41018-41027.). We have found that **Tz 1** exhibited a 98% downregulation in *c-MYC* exon 1 expression whereas the expression of exon 2 remained unchanged upon compound treatment. These results strongly demonstrate that **Tz 1** targets *c-MYC* G-quadruplex in cellular system and reduces the *c-MYC* expression.

It should be also noted that many drugs, even when approved, simultaneously target a set of signalling genes that ultimately lead to apoptosis (Expert Opin Inv Drug 2010, **19**, 1297-1307, *Bioorg. Med. Chem.* 2008, **16**, 7582-7591, *Pharmacogenet. Genom.*, 2011, **21**, 440-446.). In this regard, the compound of our interest may have other secondary effects other than targeting *c-MYC* promoter G-quadruplex, which may collectively induce apoptosis in cancer cells.

As suggested, these issues have been discussed in the revised manuscript.

Reviewer #1 (Remarks to the Author):

The authors have done a nice job responding to my comments and addressing all of the issues. My only remaining concern is whether or not the novelty of this work is sufficient for publication in Nature Communications. I will leave it up to the Editor to make the final call in that regard.

Reviewer #2 (Remarks to the Author):

Below please find my point by point reply to the rebuttal.

1) Since there is no experimental evidence for the formation of any 1,5-isomers in this study, I still think it is far-fetched to claim that the library can give rise to 66 product combinations.

2) This is answered satisfactorily in the revised version.

3) It should appear from the supporting material that the alkyne is added as a DMSO solution. See more below.

4) I agree that the positive charge in protonated 1a favors binding to the negatively charged G-quadruplex which may in part explain the selectivity. However, I still don't think you can rule out that solubility plays a role. Even though precipitation of 1b and 1c is not observed, my organic chemical experience tells me that 1b and 1c have very low solubility in a DMSO/aqueous buffer mixture and that it may just stick to the sides of the Eppendorf tube. This would also explain why products of 1b and 1c, even with the more polar azides, are not observed. Nevertheless, I will be satisfied if it is mentioned in the text that solubility may also play a role.

Response to Reviewer #2:

1) Since there is no experimental evidence for the formation of any 1,5-isomers in this study, I still think it is far-fetched to claim that the library can give rise to 66 product combinations.

In the revised manuscript, we have hypothesized in the '**Design and synthesis of azide and alkyne building blocks**' section as: "The synthesized azide and alkyne library would generate 66 theoretically possible triazole products including 1,4 and 1,5 regioisomers that may interact with the target G-quadruplex."

Since we did not observe the formation of *1,5-isomers*, as per the reviewer's suggestion, we have mentioned in the discussion section as:

"This innovative approach has been used for a model library of azide and alkyne building blocks to obtain two 1,4-triazole lead compounds (out of 33 possible triazole products of 1, 4-regiochemistry) for the G-quadruplex target".

2) It should appear from the supporting material that the alkyne is added as a DMSO solution. See more below.

-It is mentioned in the methods section in the revised manuscript.

3) I agree that the positive charge in protonated 1a favors binding to the negatively charged G-quadruplex which may in part explain the selectivity. However, I still don't think you can rule out that solubility plays a role. Even though precipitation of 1b and 1c is not observed, my organic chemical experience tells me that 1b and 1c have very low solubility in a DMSO/aqueous buffer mixture and that it may just stick to the sides of the Eppendorf tube. This would also explain why products of 1b and 1c, even with the more polar azides, are not observed. Nevertheless, I will be satisfied if it is mentioned in the text that solubility may also play a role.

-As per the suggestion, we have mentioned in the text as: "However, the solubility of these fragments in aqueous buffer may play a role in their selection."